# Paracrine regulation of neural crest EMT by placodal MMP28

**Nadège Gouignard**[1,2], **Anne Bibonne**[1], **João F. Mata**[3], **Fernanda Bajanca**[1], **Bianka Berki**[1], **Elias H. Barriga**[3], **Jean-Pierre Saint-Jeannet**[2], **Eric Theveneau**[1]*

1 Molecular Cellular and Developmental Biology department (MCD), Centre de Biologie Intégrative (CBI), Université de Toulouse, CNRS, UPS, Toulouse, France, 2 New York University, College of Dentistry, Department of Molecular Pathobiology, New York, New York, United States of America, 3 Instituto Gulbenkian de Ciência, Mechanisms of Morphogenesis Lab, Oeiras, Portugal

* eric.theveneau@univ-tlse3.fr

**Data Availability Statement:** All relevant data are within the paper and its Supporting Information files.

## Abstract

Epithelial–mesenchymal transition (EMT) is an early event in cell dissemination from epithelial tissues. EMT endows cells with migratory, and sometimes invasive, capabilities and is thus a key process in embryo morphogenesis and cancer progression. So far, matrix metalloproteinases (MMPs) have not been considered as key players in EMT but rather studied for their role in matrix remodelling in later events such as cell migration per se. Here, we used *Xenopus* neural crest cells to assess the role of MMP28 in EMT and migration in vivo. We show that a catalytically active MMP28, expressed by neighbouring placodal cells, is required for neural crest EMT and cell migration. We provide strong evidence indicating that MMP28 is imported in the nucleus of neural crest cells where it is required for normal Twist expression. Our data demonstrate that MMP28 can act as an upstream regulator of EMT in vivo raising the possibility that other MMPs might have similar early roles in various EMT-related contexts such as cancer, fibrosis, and wound healing.

## Introduction

Epithelial–mesenchymal transition (EMT) is a complex process controlled by an array of transcription factors such as members of the *snai*, *twist*, *zeb*, and *soxE* families [1,2]. During EMT, cells remodel their adhesion with other cells and the surrounding matrix and display increased cytoskeleton dynamics. These changes drive a change from an apicobasal polarity associated to epithelial stability to a front-rear polarity required for cell migration. EMT is essential for morphogenetic events such as ingression of mesodermal cells during gastrulation or emigration of neural crest cells from the neural tube but is also taking place in several diseases such as fibrosis and cancer [1–4]. EMT is an extremely complex and reversible process that is made of a series of non-obligatory steps. Therefore, despite conservation of the core changes taking place at the single-cell level, a wealth of regulatory mechanisms has been identified at the molecular level. The way cells undertake EMT appears to be highly context-dependent and renders the task of agreeing on a common definition across fields all the more challenging but common lines are starting to emerge [5].

**Funding:** This work was supported by the Fondation pour le Recherche Medicale (FRM AJE201224 to ET; ARF20150934153 to NG), the Midi-Pyrenees Regional Council (13053025 to ET), Toulouse Cancer Sante (DynaMeca to ET), the European Marie Curie Prestiges Program (PRESTIGES 2015–4–007 to ET and NG), the National Institutes of Health (R21 DE029333 to NG; R01DE25806 to JPSJ), a pilot grant from the NYU Center for Skeletal and Craniofacial Biology, which was established by NIH (1P30DE020754 to NG), the European Research Council (ERC) under the European Union's Horizon 2020 research and innovation programme (grant agreement No. 950254 to EHB)", EMBO (IG Project Number 4765 to EHB) and la Caixa (Junior Leader Incoming 94978 to EHB). ET receives his salary from the French National Center for Scientific Research (CNRS). The funders had no role in study design, data collection and analysis, decision to publish, or preparation of the manuscript.

**Competing interests:** The authors have declared that no competing interests exist.

**Abbreviations:** ChIP, chromatin immunoprecipitation; CMO, control Morpholino; EMT, epithelial–mesenchymal transition; HCC, hepatocellular carcinoma; ISH, in situ hybridization; MMP, matrix metalloproteinase; NES, nuclear export signal; NLS, nuclear localization signal; ODC, ornithine decarboxylase.

Matrix metalloproteinases (MMPs) are secreted enzymes initially discovered for their ability to remodel the extracellular matrix [6] and early evidence showed that MMPs could influence EMT via their role on the extracellular space [7–9]. Somehow the link between EMT and MMPs was never fully explored. This leads to the current situation where MMPs are not considered as relevant markers or regulators of EMT [5]. However, we now know that MMPs are pleiotropic players in health and diseases that can influence growth, survival, and migration [10]. MMPs have numerous noncanonical subcellular localizations (e.g., mitochondria, nucleus, cytoplasm) and several unexpected substrates have been described (e.g., cell adhesion molecules, growth factors, guidance cues) [6,11,12]. These observations suggest numerous putative functions that are not related to the regulation of the extracellular matrix but the functional and physiological relevance of these potential noncanonical functions still awaits demonstration. In particular, it is interesting to note that most MMPs have been detected in the nucleus [13] of at least 1 cell type and that some have been shown to exhibit transcriptional roles and DNA-binding abilities [14–16]. Given the frequent expression of MMPs by cells undergoing EMT, this calls for a re-assessment of their involvement in EMT independently of their effects on extracellular matrix.

Here, we used *Xenopus* neural crest cells to assess the putative role of MMP28 in EMT. MMP28 is the last member of the MMP family to have been identified in human. It has a typical MMP structure with a secretion signal, a pro-domain that needs to be removed for complete enzymatic activity, and a hemopexin-like domain involved in cofactors binding and substrates recognition [17]. Roles and functions of MMP28 are poorly documented as compared to other members of the family. It has been shown to be involved in wound healing and nerve repair [17]. It is expressed in pulmonary fibrosis [18] and several human cancers including gastric cancer where it correlates with poor prognosis [19].

Neural crest cells are multipotent stem cells that form at the interface between the neural and nonneural ectoderm [20]. They perform EMT to initiate cell migration and go on to colonize most tissues and organs of the developing embryo [21]. Neural crest EMT relies on oncogenes such as *snai2* and *twist* [22]. The neural crest EMT program is often hijacked by invasive cells during carcinoma progression [23,24], making these cells an extremely relevant in vivo model to study EMT. Our data show that, during *Xenopus* development, MMP28 expressed in cranial placodes [25] is required for EMT of neural crest cells. MMP28 is secreted by placode cells, imported into the nucleus of adjacent neural crest cells where its catalytic activity is required for proper implementation of EMT via the maintenance of *twist* expression.

Overall, our results demonstrate a paracrine role for MMP28 in the EMT program of neural crest cells in vivo suggesting that such paracrine role might take place between other cells expressing MMPs such as fibroblasts and cancer cells.

## Results

### MMP28 is required for the expression of multiple neural crest genes

We found that MMP28, a secreted metalloproteinase, is expressed in cranial placodes adjacent to the cephalic neural crest [25] and given the known importance of neural crest interaction with placodes for normal neural crest migration [26]; we decided to assess the putative role of placodal MMP28 for neural crest development. MMP28 expression starts before the onset of neural crest migration (Fig 1A and 1B) and comparison with markers for neural crest (*sox8* and *snai2*), neural plate (*sox2*), and placodes (*six1*) (Fig 1B and 1C) confirms that MMP28 expression is restricted to the posterior part of the pan-placodal domain. In addition, a small neural crest domain, organised as a thin line along the edge of the neural fold, called the medial crest, also expressed MMP28 as we previously described [25]. MMP28 expression is then

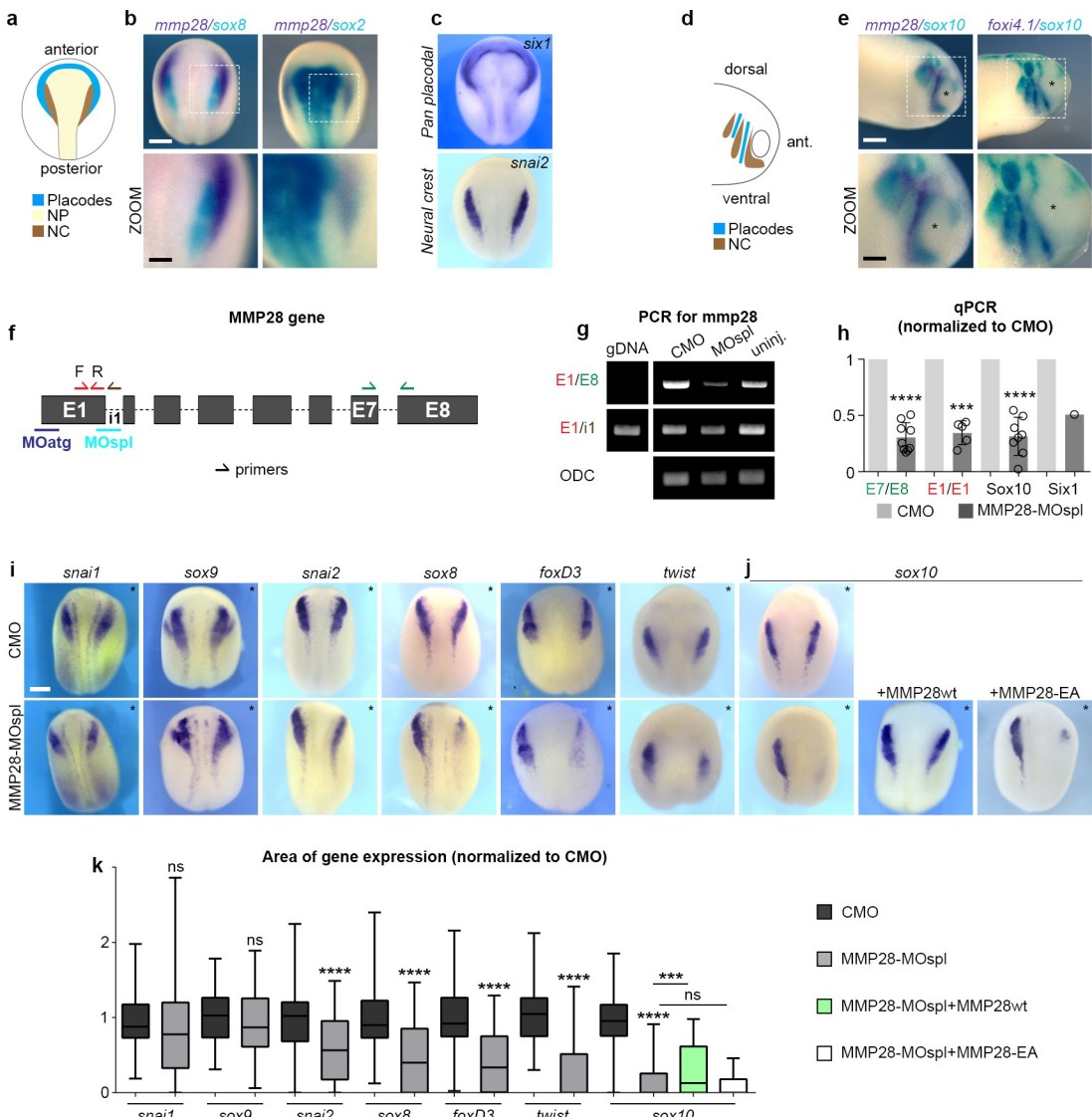

**Fig 1. MMP28 expressed in placodes is required for normal expression of neural crest genes.** (a) Diagram depicting the distribution of placodes, neural crest (NC) and neural plate (NP) at stage 16. (b, c) In situ hybridization for *mmp28*, *sox2* (neural plate), *sox8* (neural crest), *six1* (placodes), and *snai2* (neural crest), as indicated. (d) Diagram depicting the distribution of placodes and neural crest at stage 25. (e) In situ hybridization for *mmp28*, *sox10* (neural crest), and *foxi4.1* (epibranchial placodes), as indicated. (f) Diagram of mmp28 gene organisation and the relative positions of MMP28 anti-splicing (MOspl) and translation-blocking Morpholinos (MOatg) used in this study as well as the position and orientation of primers for PCR used to assess MOspl efficiency. (g) Result of PCR with the various combinations of primers shown in (f). ODC is used as loading control. (h) Quantitative PCR with the various combinations of primers shown in (f), normalised to control Morpholino (CMO); replicates $n_{E7/8}$ = 9; $n_{E1/1}$ = 5; $n_{sox10}$ = 8; Six1 was done only once. One-sample *t* test against a theoretical values of 1, **** $p < 0.0001$, *** $p = 0.002$. Small arrows indicate the position and orientation of primers. All sequences are in the Materials and methods section. (i, j) Phenotype of embryos (stage 16) injected with CMO or MMP28 Morpholino (MMP28-MOspl) alone or in combination with wild-type (wt) or catalytically dead mutant (EA) MMP28 mRNA and analysed by in situ hybridization for neural crest genes *snai1*, *sox9*, *snai2*, *sox8*, *foxd3*, *twist*, or *sox10* expression (as indicated), asterisks indicate injected side. (k) Area of expression of neural crest genes normalised to the non-injected side and CMO condition analysed from 6 independent experiments. Number of embryos per condition, from left to right: 47, 72; 48, 36; 63, 62; 90, 53; 43, 65, 25, 64; 100, 165, 52, 75. Unpaired *t* test with Welch's correction (CMO vs. MMP28-MO) for all genes except *sox10*. For *sox10*, ANOVA followed by multiple comparisons, ns $p > 0.1550$, *(MMP28-MO vs. MMP28-MO+wt) $p = 0.0292$, ****(CMO vs. MMP28-MO) $p < 0.0001$. Scale bar, panels (b), (c), (e), and (i), 250 μm; zooms, 100 μm. Numerical data from all graphs can be found in the supporting S1 Data file. Raw images of gels and blots can be found in the S1 Raw Images file. MMP, matrix metalloproteinase; NC, neural crest; NP, neural plate; ODC, ornithine decarboxylase.

maintained in placodes at neural crest migration stages (Fig 1D and 1E). For a complete description of the early expression pattern of MMP28 in *Xenopus laevis*, please refer to ref [25]. To assess its functional relevance, we performed loss-of-function experiments using a splice blocking Morpholino directed against MMP28 (MMP28-MOspl), whose efficiency was assessed by RT-PCR and qPCR (Fig 1F–1H). Knocking down MMP28 led to a severe down-regulation of multiple neural crest genes including *twist*, *sox10*, *snai2*, *sox8*, and *foxd3*, whereas other genes such as *snai1* and *sox9* were not affected (Fig 1I–1K). To assess whether some of these effects might be due to the aforementioned MMP28 expression in the medial crest, we performed MMP28 knockdown in neural plate and the medial crest, but not the placodes, by targeted injections of dorsal blastomeres and found no effect on *sox10* expression (S1 Fig).

Importantly, MMP28 knockdown had no effect on nonneural ectoderm, neural plate, or neural plate border gene expression and some effects on placodes themselves with a reduction of *six1* expression but without affecting *eya1* or *foxi4.1* expressions (S2 Fig). This indicates that, while MMP28 is required for normal expression of multiple neural crest genes, the neural crest territory is still induced and properly positioned in absence of MMP28 and can be identified by the co-expression of *sox9* and *snai1*. To further substantiate this, we performed a TUNEL assay and found no induction of cell death after injection of the control Morpholino (CMO) or MMP28spl-MO (S3 Fig) compared to the inhibition of Sf3b4, an mRNA splicing factor, which is known to specifically promote cell death in neural crest cells [27].

*Sox10* being the most affected neural crest gene, we attempted to rescue its expression by co-injecting MMP28-MOspl together with wild-type MMP28 (MMP28wt) or a previously described [17] inactive point mutant version in which the catalytic activity is abolished (MMP28-EA). MMP28wt was sufficient to rescue *sox10* expression in morphant embryos (Fig 1J and 1K, green bar), whereas MMP28-EA was not (Fig 1J and 1K, white bar). Importantly, MMP28wt and MMP28-EA do not have a dominant-negative effect (S4 Fig). This indicates that their respective abilities to rescue MMP28 loss-of-function cannot be explained by putative interference with other MMPs. Furthermore, MMP28wt-GFP rescues expression of Sox10 in morphant embryos but not that of Six1 indicating that the role of MMP28 in neural crest cells and placodes can be uncoupled (S5 Fig) and in particular that rescue of placodal gene expression is not a prerequisite for that of neural crest genes.

Therefore, these data show that MMP28 secreted by placodes is dispensable for the formation and positioning of the neural crest territory but that its catalytic activity is specifically required for the normal expression of multiple neural crest genes prior to neural crest migration.

## MMP28 is required for normal neural crest EMT and migration

Since inhibition of MMP28 led to a reduction of important EMT regulators such as *twist* and *snai2*, we wanted to evaluate whether this translated into actual EMT and migration defects. First, we assessed the dorsoventral extension of neural crest streams after MMP28 knockdown with 2 independent Morpholinos (MOspl, MOatg). MOatg is a translation blocking MO that is designed to bind upstream of the ATG of MMP28 (see Materials and methods). We found a severe reduction of the net distance migrated by neural crest cells with both treatments compared to CMO (Fig 2A and 2B).

Second, we performed ex vivo neural crest culture [28] followed by time-lapse imaging to monitor cell dispersion (Fig 2C–2E and S1 Movie). Neural crest cells taken from embryos injected with CMO extensively moved away from their initial position (Fig 2C–2E, black graph and curve), whereas cells coming from embryos in which MMP28 was knocked down (MOatg) failed to disperse (Fig 2C–2E, brown graph and curve, and S1 Movie). Importantly,

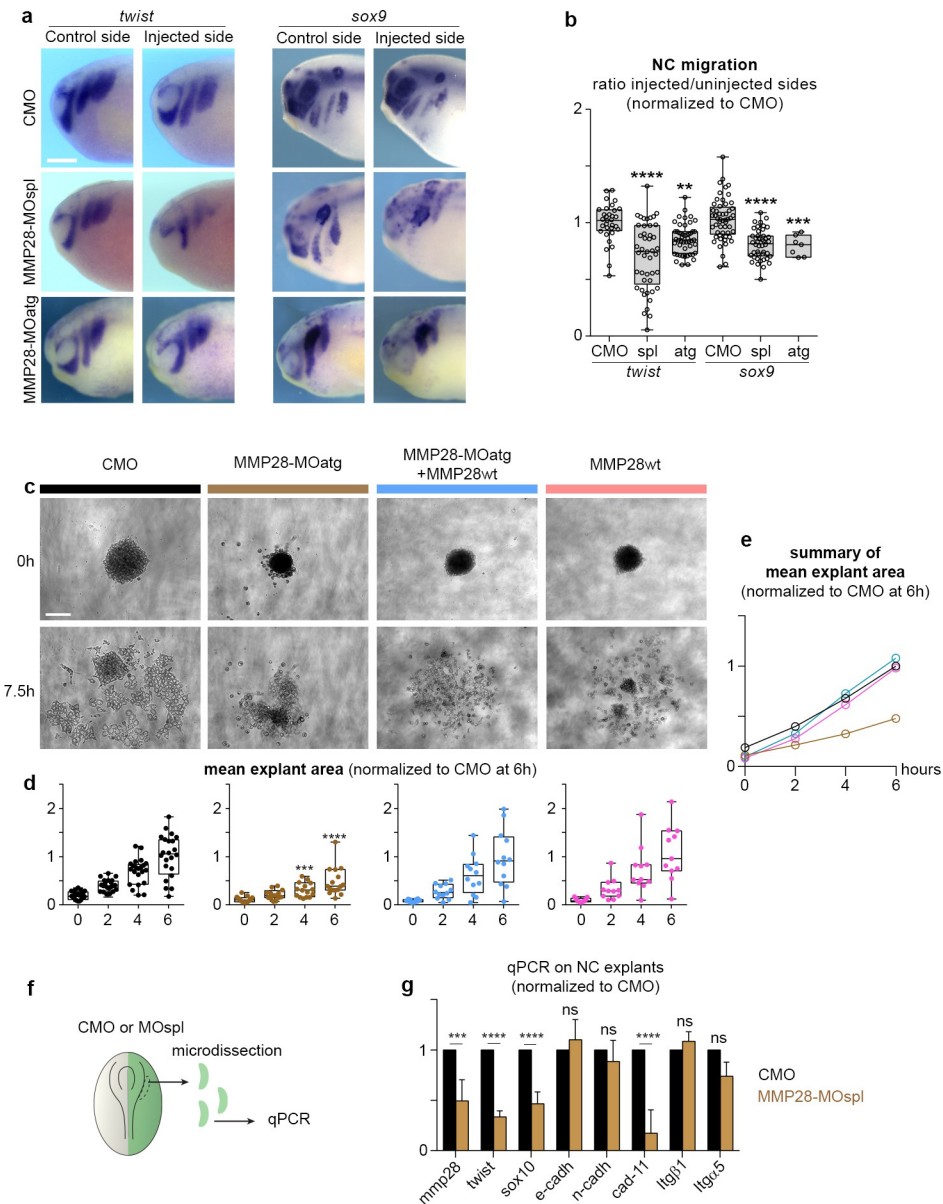

**Fig 2. MMP28 is required for normal EMT and cell migration.** (a) Phenotype of CMO, MMP28-MOspl, and MMP28-MOatg-injected embryos (stage 25) analysed for *twist* and *sox9* expression, scale bar, 250 μm. (b) Graph plotting the distance migrated by NC cells in CMO and MMP28-MO embryos. t*wist*: nCMO = 32, nMOspl = 45, nMOatg = 49; *sox9*: nCMO = 53, nMOspl = 44, nMOatg = 7. ANOVA followed by multiple comparisons, **** $p < 0.0001$, ***, $p = 0.0005$; **, $p = 0.0034$. (c) Representative examples of explants at t0 (1 h after plating on fibronectin) and +7.5 h, scale bar, 100 μm. (d, e) Distribution of explant areas per hour for CMO ($n = 24$), MMP28-MOatg-8ng ($n = 16$), MOatg-8ng+MMP28wt-1200pg ($n = 11$), MMP28wt-1200pg ($n = 12$). Analysed by two-way ANOVA per time point, ***, $p = 0.0003$; ****, $p < 0.0001$. (f) Diagram depicting the procedure prior to quantitative PCR. (g) Quantitative PCR for expression of *mmp28*, *sox10*, *twist*, *cadherins E, N* and *11*, *integrin α5*, and *β1* subunits, after injection of MMP28-MOspl or CMO. The values are normalised to *eef1a1* and to the levels of expression in CMO. From 3 independent mRNA extractions. Two-way ANOVA, *p* values CMO vs. MMP28-MO: *mmp28* = 0.0001 (***), *sox10* < 0.0001 (****), *twist* < 0.0001, *e-cadherin* 0.9498 (ns), *n-cadherin* 0.8943 (ns), *cadherin-11* < 0.0001 (****), *integrin-β1* 0.9813 (ns), *integrin-α5* 0.0915 (ns). Numerical data from all graphs can be found in the supporting S1 Data file. CMO, control Morpholino; EMT, epithelial–mesenchymal transition; MMP, matrix metalloproteinase; NC, neural crest.

the effect of knocking down MMP28 with MOatg can be rescued by expressing MMP28wt, which does not contain the binding site of MOatg, (Fig 2C–2E, blue graph and curve, and S2 Movie). By contrast, overexpressing MMP28wt has no visible effect ex vivo (Fig 2C–2E, pink graph and curve, and S2 Movie) in line with the lack of effect observed in vivo (S4 Fig).

Third, we monitored the relative expression of cell adhesion molecules. To initiate migration, *Xenopus* neural crest cells down-regulate the expression of E-cadherin and up-regulate that of N-cadherin and cadherin-11 to perform contact-inhibition of locomotion [29–31] and rely on α5β1 integrins and cadherin-11 to bind to Fibronectin [31,32]. Therefore, we assessed the expression of these genes in neural crest by qPCR after MMP28 knockdown, using the expression of *mmp28*, *sox10*, and *twist* as internal controls for the MMP28-MOspl efficiency (Fig 2F and 2G). Injection of the MMP28-MOspl reduced the expression of *mmp28*, *twist*, and *sox10* by half confirming the efficiency of the knockdown in these samples. By contrast, MMP28 knockdown had no significant effects on E- and N-cadherins compared to embryos injected with the CMO indicating that embryos with MMP28 knockdown have the expected low E-cadherin/high N-cadherin profile. By contrast, MMP28 knockdown severely reduced the expression of cadherin-11. Finally, integrin subunits α5 and β1 were not affected (Fig 2G). This indicates that MMP28-MO neural crest cells initiate EMT but fail to complete it. Altogether, these 3 independent analyses show that the inhibition of EMT transcription factors under MMP28 knockdown conditions translates into EMT and migration defects at tissue (Fig 2A and 2B), cellular (Fig 2C–2F), and molecular (Fig 2F and 2G) levels.

## Twist expression is sufficient to rescue neural crest adhesion and migration in MMP28 knockdown embryos

Given that the expressions of *twist*, an upstream regulator of EMT, and *cadherin-11*, a downstream effector of the EMT cascade, are severely affected by MMP28 knockdown, we wondered whether forcing expression of either one of these genes might be sufficient to rescue the MMP28 knockdown phenotype. For that, we co-injected MMP28-MOspl with the mRNA for *twist* or *cadherin-11* and analysed neural crest migration using *foxd3* expression (Fig 3A–3C). Embryos injected with the CMO displayed normal neural crest migration on both the uninjected and the injected sides, while embryos injected with MMP28-MOspl had impaired neural crest migration on the injected side. Interestingly, co-injection with *twist* mRNA was sufficient to partially restore dorsoventral neural crest migration while co-injection of *cadherin-11* mRNA was not (Fig 3B and 3C).

To better understand how these treatments affected neural crest cells, we plated the various conditions onto Fibronectin. First, we let explants migrate for 3 h and fixed them for nuclear and actin staining with DAPI and Phalloidin, respectively (Fig 3D–3F). The explants that were still attached after fixation were counted, washed, and then stained. Fig 3E shows a low magnification of each well with adhering explants outlined in purple and detached explants outlined in yellow. In CMO conditions and MMP28-MO+*twist* mRNA, all explants remained attached whereas after MMP28 knockdown only 5 out of 12 explants had a significant amount of cells left attached to the dish. Interestingly, in the MMP28-MO+*cadherin-11* mRNA, 8 out of 10 explants remained attached to the substrate (Fig 3E and 3F). We next looked at cell morphology. Cells injected with CMO or MMP28-MO together with *twist* or *cadherin-11* were able to flatten on the substrate and to form protrusions. By contrast, MMP28-MO cells were mostly round and poorly protrusive (Fig 3G). Importantly, the nuclear staining did not reveal any fragmented nuclei, in line with our in vivo TUNEL data (S3 Fig). As a proxy for membrane dynamics, we looked at the size of protrusions in each condition. We measured the area of protrusions either directed toward a free space (Fig 3H, outward protrusions) or in between cells

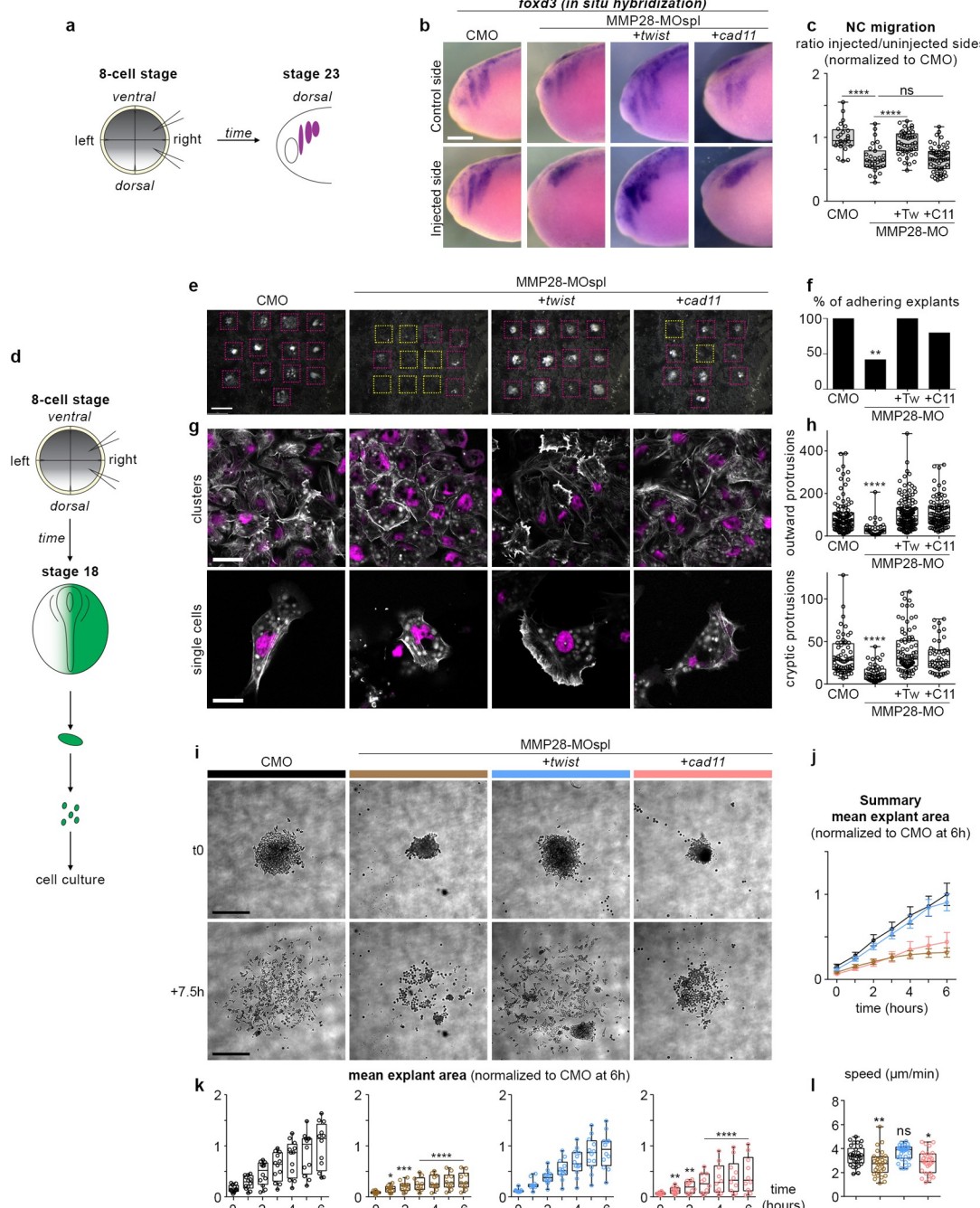

**Fig 3. Twist expression is sufficient to rescue adhesion and migration of neural crest cells after MMP28 knockdown.** (a) Diagram depicting the experimental set-up with injection of MO and mRNA in 2 blastomeres on one side of 8-cell stage embryos and the embryos analysed at neural crest migration stage (stage 23). (b) In situ hybridization against *foxd3* following injection with CMO, MMP28-MOspl, or the co-injection of MMP28-MOspl together with *twist* or *cadherin-11* mRNA. (c) Graph plotting the distance migrated by neural crest cells in the experimental conditions shown in (b), nCMO = 25, nMMP28-MOspl = 31 nMMP28-MOspl+*twist* = 57, nMMP28-MOspl+*cad11* = 47 from 2 independent experiments. ANOVA followed by uncorrected Fisher's LSD; ****, *p* < 0.0001, ns, *p* = 0.8829. (d) Diagram depicting the experimental procedure for neural crest culture on Fibronectin. (e) Low magnification images of explants in all experimental conditions after fixation. Adhering explants are outlined in purple, detached explants are outlined in yellow; scale bar, 500 μm. (f) Quantification of adhering explants, CMO = 13/13; MMP28-MOspl = 5/13; MMP28-MOspl+*twist* = 12/12; MO28-MOspl+*cad11* = 8/10. Contingency tables for the comparison of proportion; CMO vs. MMP28-MOspl, T = 10.53, alpha 0.01 (**), MMP28-MOspl vs. MMP28-MOspl+*twist*, T = 9.88, alpha 0.01 (**), MMP28-MOspl vs. MMP28-MOspl+*cad11*, T = 3.31 (ns), CMO vs. MMP28-MOspl+*cad11*, T = 2.85, (ns). (g) DAPI (magenta) and Phalloidin (white) staining; scale bars, 40 μm for clusters,

20 μm for single cells. (h) Protrusion area in μm$^2$, outward protrusions CMO ($n = 64$), MMP28-MOspl ($n = 49$), MOspl+*twist* ($n = 47$), MOspl+*cad11* ($n = 87$); cryptic protrusions CMO ($n = 103$), MMP28-MOspl ($n = 29$), MOspl+*twist* ($n = 140$), MOspl +Cad11 ($n = 100$); ANOVA, Kruskal–Wallis test; ****, $p < 0.0001$. (i) Time-lapse imaging of neural crest explants, scale bar, 250 μm. (j) Summary graph with curves showing the mean area and standard deviation per experimental condition shown in (i) per time point. (j, k) Mean explant area + SD per explant per conditions shown in (i) per time point, CMO ($n = 12$), MMP28-MOspl ($n = 10$), MOspl+*twist* ($n = 12$), MOspl+*cad11* ($n = 10$). (l) Speed of individual cells (μm/min) from each experimental conditions shown in panels (i–k); CMO ($n = 30$), MMP28-MOspl ($n = 30$), MOspl+*twist* ($n = 32$), MOspl+*cad11* ($n = 28$). ANOVA followed by multiple comparisons; CMO vs. MOspl $p = 0.0057$ (**), CMO vs. MOspl+*cadherin11* $p = 0.0291$ (*), CMO vs. MOspl+*twist* $p = 0.2483$ (ns). Numerical data from all graphs can be found in the supporting S1 Data file. CMO, control Morpholino; MMP, matrix metalloproteinase.

(Fig 3H, cryptic protrusions). *Twist* and *cadherin-11* mRNA were both sufficient to rescue protrusive activity in MMP28-MO cells (Fig 3H).

Next, we assessed cells dynamics by time-lapse imaging (Fig 3I–3K and S3 Movie). Cells injected with CMO dispersed normally (Fig 3I–3K, black graphs and curves) while MMP28-MO cells were round and failed to significantly disperse (Fig 3I–3K, brown graphs and curves). Interestingly, expression of *twist* was able to restore normal dispersion of MMP28-MO injected neural crest cells (Fig 3I–3K, blue graphs and curves) while expression of *cadherin-11* was unable to do so (Fig 3I–3K, pink graphs and curves). This is further substantiated by tracking of single cells located at the border of explants under each experimental condition showing that cells coming from embryos injected with MMP28-MO or MMP28-MO + *cadherin-11* have a significantly lower speed than cells from control or MMP28-MO + *twist* embryos (Fig 3I). Altogether, these data indicate that Twist and Cadherin-11 are essential players downstream of MMP28. Cadherin-11 is sufficient to restore cell-matrix adhesion but this rescue of adhesion does not translate into an efficient rescue of cell motility in or ex vivo. By contrast, Twist expression fully restores adhesion and dispersion ex vivo and significantly restores neural crest migration in vivo.

## MMP28 can traffic to the nucleus of neural crest cells

Next, we wondered how MMP28 might affect the expression of neural crest genes while being produced by adjacent placodal cells. Despite being secreted, several MMPs, including MMP28, have been detected in the nucleus of various cell types [13,18]. Thus, we analysed the amino acid sequence of *Xenopus* MMP28 for putative nuclear export and nuclear localization signals (NES/NLS). We found 2 putative NES and 2 putative NLS sites in MMP28 (Fig 4A). To test whether MMP28 is able to traffic to the nucleus, we expressed a GFP-tagged version of MMP28 in *Xenopus* embryos, harvested embryos and performed cell fractionation followed by western blot (Fig 4B). We found MMP28 in the membrane (Mem), the soluble cytosol (Cy) and the cytoskeleton-associated (CytoSK) fractions as well as the soluble (Sol) and chromatin-associated (Chr) nuclear fractions (Fig 4B). To assess whether the MMP28 detected in the nuclear fractions might be due to contamination from the cytosolic fractions, we run the cytosolic and nuclear fractions from uninjected and embryos expressing MMP28-GFP in parallel and performed the immunoblots with anti-GFP (Fig 4C) and anti-tubulin (Fig 4D). No tubulin was found in nuclear extracts (Fig 4D) while MMP28-GFP was clearly detected (Fig 4C). This shows that the detection of MMP28 in nuclear fractions cannot be explained by a contamination from the cytosolic fraction.

To substantiate these data, we expressed MMP28-GFP in neural crest cells and performed anti-GFP immunostaining followed by 3D confocal imaging (Fig 4E and S4 Movie). Nuclei were counterstained with DAPI and the DAPI signal used as a mask to probe the amount of GFP signal embedded within it. Multiple GFP spots were detected in the nuclei of neural crest cells (Fig 4E, zooms). To assess whether the addition of the GFP-tag might be responsible for a

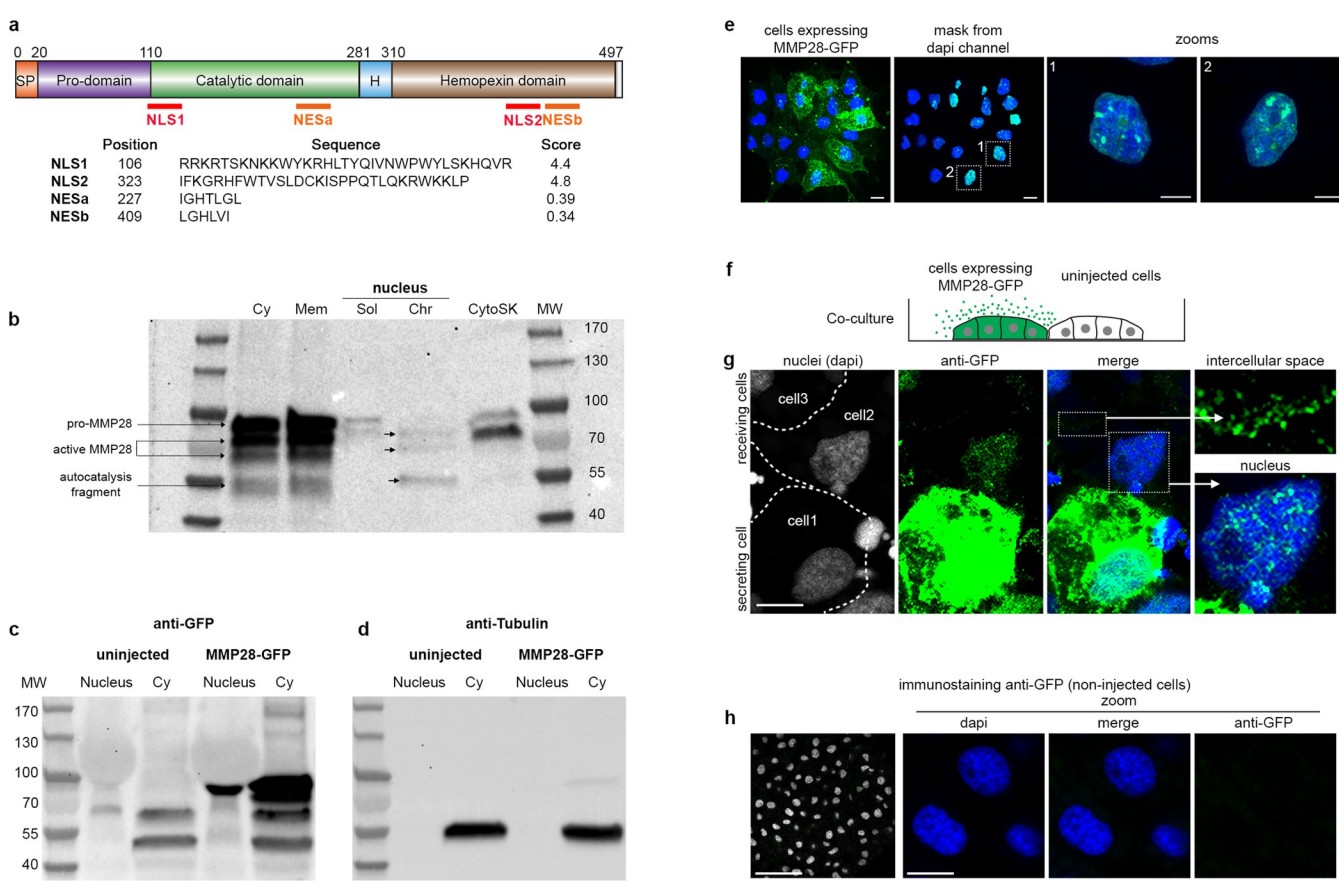

**Fig 4. MMP28 can be imported in the nucleus in a paracrine manner.** (a) Diagram of MMP28 protein structure and relative position, sequence and score of the putative NLS and NES signals identified by bioinformatics (see Materials and methods). SP, signal peptide; H, hinge region. (b) Western blot against GFP after cell fractionation from embryos expressing MMP28-GFP; Cy, cytosol; Mem, membrane; Sol, soluble nuclear fraction; Chr, chromatin-bound nuclear fraction; CytoSK, cytoskeleton fraction; MW, molecular weight. (c, d) Western bots against GFP (c) and tubulin (d) on nuclear and cytosolic fractions. (e) Neural crest explant expressing MMP28-GFP (green), counterstained with DAPI (blue) observed by 3D confocal imaging, scale bar, 10 μm, dash line squares indicate zoomed areas. (f) Diagram depicting the co-culture assay. (g) Control *Xenopus* neural crest cells and neural crest cells expressing MMP28-GFP co-cultured on Fibronectin, immunostained with anti-GFP antibody (green) and counterstained with DAPI (blue), representative images from 2 independent experiments with 6 explants, scale bar, 10 μm. (h) Immunostaining against GFP on non-injected cells. Raw images of gels and blots can be found in the S1 Raw Images file. MMP, matrix metalloproteinase; NES, nuclear export signal; NLS, nuclear localization signal.

nonspecific nuclear accumulation of MMP28, we performed the same experiment with a flag-tagged version of MMP28. MMP28-flag was also detected in the neural crest nuclei excluding the possibility that the GFP tag nonspecifically drives MMP28 nuclear accumulation (S5 Movie).

In vivo, MMP28 is not expressed by neural crest cells but by nearby placodal cells. Therefore, to test whether MMP28 can traffic to neighbouring cells in a paracrine manner, we co-cultured control neural crest cells and cells expressing MMP28-GFP and performed anti-GFP immunostaining. We detected MMP28-GFP in the intercellular space between expressing and non-expressing cells as well as in the nucleus of non-expressing cells (Fig 4F and 4G). To assess the specificity of the GFP immunostaining, we performed GFP immunodetection on uninjected cells and did not detect any significant staining under the same confocal conditions (Fig 4H).

## MMP28 produced by placodal cells can be imported into neural crest cells' nuclei in vivo

We next tested whether MMP28 could travel from placodes to neural crest cells in vivo within a time window compatible with normal neural crest-placodes interactions. To do so, we expressed MMP28wt-GFP, MMP28-EA-GFP, or a secreted form of GFP containing the signal peptide of MMP28 as a control, in the ectoderm of *Xenopus* embryos, and grafted neural crest explants labelled with rhodamine-dextran as a tracer next to the placodal region (Fig 5A). Embryos were fixed 4 h after the graft and processed for histology and confocal imaging to monitor the raw GFP signal. MMP28-GFP was detected in the cytoplasm and the nucleus of multiple neural crest cells located underneath the placodal ectoderm expressing MMP28wt-GFP or MMP28-EA-GFP (Fig 5B–5D). By contrast, neural crest cells grafted near the placodal ectoderm expressing secreted GFP had no GFP signal in their cytoplasm or nuclei, showing that GFP alone is not spontaneously endocytosed or imported in the nucleus. These data indicate that MMP28 is specifically imported and that the catalytic activity is not required for the import. To get a broader view of the distribution of MMP28-GFP in grafted neural crest cells, we performed similar grafts with MMP28wt-GFP followed by immunodetection of GFP (S6 Fig). MMP28-GFP is not restricted to cells directly underneath the MMP28-expressing placodal ectoderm and is found up to several cell diameters away from the ectoderm. By contrast, immunodetection against GFP on uninjected embryos led to no significant signals confirming the specificity of the observed staining (S6 Fig). These data show that MMP28 can travel from the placodal ectoderm to the nuclei of neural crest cells within a few hours in vivo.

Is MMP28 import depending on the neural crest-placodes interactions? To assess this, we designed several grafting experiments. We placed neural crest explants labelled with rhodamine-dextran next to caudal ectoderm producing MMP28-GFP to test MMP28 import from a non-placodal source. We grafted placodes labelled with rhodamine-dextran next to endogenous placodes producing MMP28-GFP to test whether placodes can also import MMP28. Finally, we made sandwiches of animal cap explants that are non-specified epithelia that have not yet acquired neural, neural crest, or placodal identities to further test for the universality of MMP28 import. A small animal cap explant labelled with rhodamine-dextran is engulfed in a bigger one producing MMP28-GFP (S7 Fig). In all conditions presented in Figs 5 and S7, we quantified the percentage of cells with internalised MMP28 in the cytoplasm and the nucleus (Fig 5C). We statistically compared the proportions of cells with MMP28 taken up to the cytoplasm and the nucleus between in each grafted condition to the control situation where neural crest cells receive MMP28-GFP from placodes (Fig 5D). These data show that the most favourable conditions to observe nuclear MMP28 are when neural crest cells are exposed to placodes expressing MMP28-GFP, regardless of MMP28 activity (Fig 5C and 5D, brown and black closed circles). However, internalisation and nuclear import can also be seen in neural crest cells exposed to MMP28-GFP produced by the caudal ectoderm and in placodes exposed to producing placodes, albeit at a lower efficiency (Fig 5C and 5D, open circle and open square). In addition, these experiments show that the ability to internalise MMP28-GFP does not predict its subsequent nuclear import. Indeed, the animal cap sandwiches are the experimental condition that leads to the highest rate of internalisation (Fig 5C, black cross) while having one of the lowest rates of nuclear import. In conclusion, internalisation of MMP28 in a paracrine manner is not specific to the neural crest–placodes interaction but the rate of nuclear import is higher when MMP28 is produced by placodes and received by neural crest cells than in any of the other experimental conditions tested.

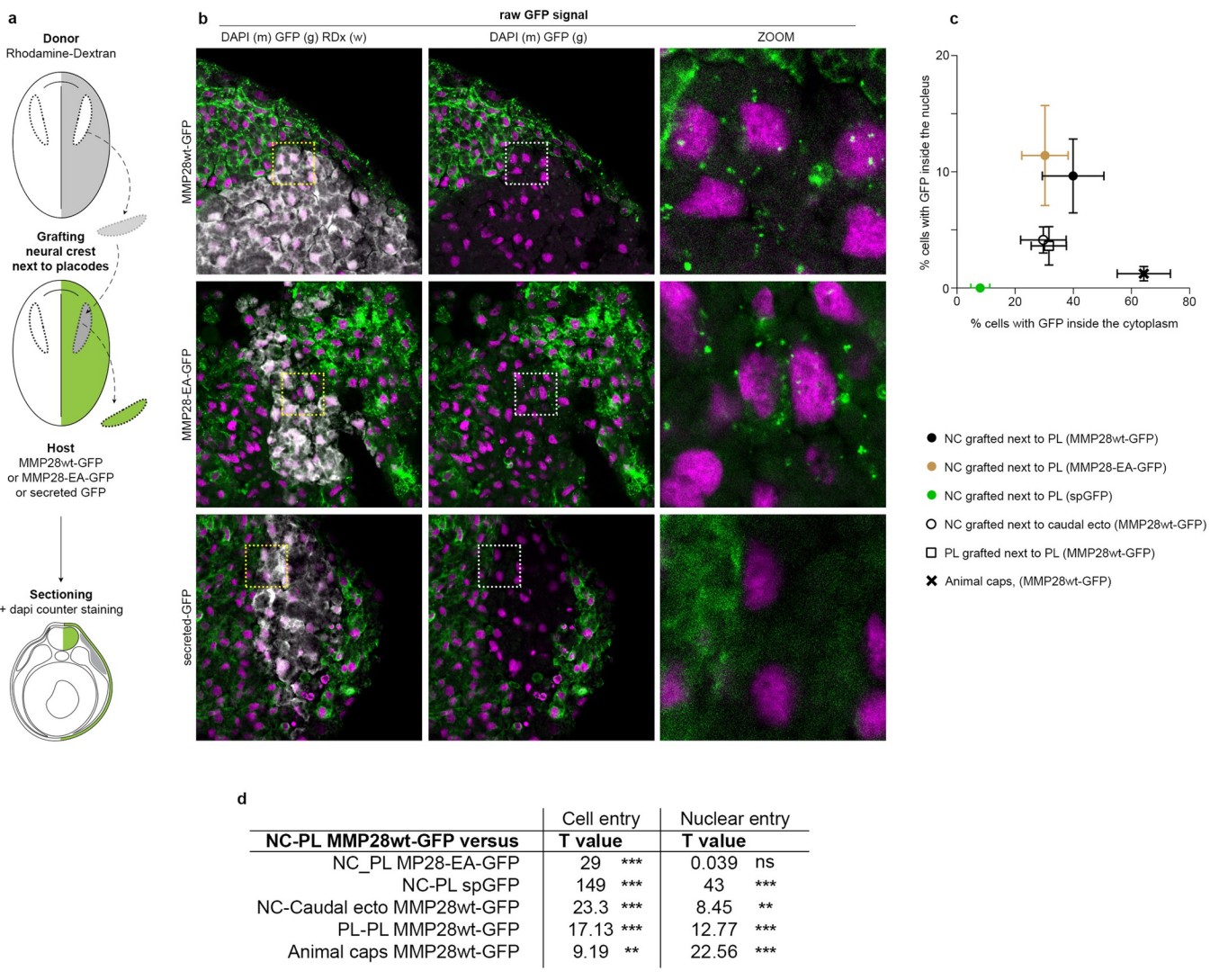

**d**

| NC-PL MMP28wt-GFP versus | Cell entry | | Nuclear entry | |
|---|---|---|---|---|
| | T value | | T value | |
| NC_PL MP28-EA-GFP | 29 | *** | 0.039 | ns |
| NC-PL spGFP | 149 | *** | 43 | *** |
| NC-Caudal ecto MMP28wt-GFP | 23.3 | *** | 8.45 | ** |
| PL-PL MMP28wt-GFP | 17.13 | *** | 12.77 | *** |
| Animal caps MMP28wt-GFP | 9.19 | ** | 22.56 | *** |

**Fig 5. MMP28 can travel from the ectoderm to the nucleus of neural crest cells in vivo.** (a) Diagram depicting the grafting procedure and sample preparation. Neural crest from a donor embryo labelled with rhodamine-dextran (grey) were grafted into a host embryo expressing MMP28wt-GFP, MMP28-EA-GFP, or secreted-GFP (green) in the ectoderm and processed for imaging. (b) Representative 1 μm-thick optical sections through the grafted area by confocal microscopy for each condition counterstained with DAPI (magenta), scale bar, 100 μm. Dash line squares indicate zoomed areas, scale bar for zooms 10 μm. No anti-GFP immunostaining was performed on these samples. (c, d) Quantification of cell internalisation and nuclear detection of MMP28-GFP in each grafted conditions shown in panels Figs 5a and 5b and S7. NC-PL MMP28wt-GFP 4 grafts, 452 cells; NC-PL MMP28-EA-GFP 2 grafts, 398 cells; NC-PL sp-GFP 4 grafts, 419 cells; NC-caudal ectoderm 7 grafts, 419 cells; PL-PL-MMP28wt-GFP 5 grafts, 259 cells; animal caps 7 sandwiches, 283 cells. Percentages of GFP-positive cells and nuclei were calculated per embryos. Means and standard deviation are plotted per experimental condition. Statistical analyses (d) comparing the proportions of cells with internal/nuclear GFP signal between NC-PL MMP28wt-GFP condition and the other conditions using contingency tables (see Materials and methods). Numerical data from all graphs can be found in the supporting S1 Data file. MMP, matrix metalloproteinase; NC, neural crest; PL, placodes.

## Active MMP28 is required within the nuclei of neural crest cells for normal *twist* expression

To show that MMP28 can travel to the nucleus of neural crest cells does not demonstrate that the nuclear localization per se is required for its role in neural crest development. MMP28 being secreted from the placodes, it is likely that MMP28 might act at multiple locations along the paracrine route: the extracellular space, the cytoplasm, and the nucleus. To assess that, we designed versions of MMP28 that would either be prevented from accumulating in the

nucleus, by adding a strong NES signal or be sequestered in the nucleus. For the later, we added a strong NLS. In addition, we also removed the secretion peptide (ΔSP) so that the requirement of the extracellular localization could be assessed. We confirmed that the MMP28$^{NES}$ and MMP28$^{ΔSPNLS}$ localised to the expected cellular compartments (Fig 6A and 6B). Since MMPs are usually activated by removal of their pro-domain while passing through the Golgi [33], we assessed whether deletion of the secretion peptide might affect removal of the pro-domain. We compared the profiles of MMP28wt, MMP28$^{ΔSP}$, and MMP28$^{ΔSP/NLS}$ in the soluble and chromatin-associated nuclear fractions by western blot. Indeed, preventing entry into the secretion pathway inhibited removal of the pro-domain (S8 Fig). We then used the NES and ΔSPNLS versions of MMP28 to attempt to rescue MMP28 knockdown in vivo. MMP28$^{NES}$, which contains the normal signal peptide, was not able to rescue *sox10* or *twist* expression. By contrast, the non-secreted nuclear-targeted ΔSPNLS version of MMP28 was able to do so (Fig 6C and 6D).

Given that the ΔSPNLS form still has its pro-domain (S8 Fig), this result suggests that the catalytic activity may not be required for MMP28 function in the nucleus. However, pro-MMPs have basal catalytic activity and can be further activated by a change of conformation known as an allosteric activation [33,34]. Therefore, the rescue obtained with the ΔSPNLS form is not a definitive proof that catalytic activity is dispensable in the nucleus. To address this point, we generated a catalytically inactive version of the ΔSPNLS (MMP28$^{EA/ΔSPNLS}$). Importantly, this catalytically inactive form of nuclear MMP28 was not able to rescue MMP28 knockdown (Fig 6C and 6D) indicating that the catalytic activity of MMP28 is indeed required within the nucleus.

## MMP28 interacts with some domains of *twist* and *cadherin-11* promoters

Finally, we wondered whether MMP28 might interact with the regulatory sequences of some of the down-regulated neural crest genes. To assess that, we performed a chromatin immuno-precipitation (ChIP) assays followed by PCR (ChIP-PCR) against multiple portions of the promoters of *sox10*, *cad11*, and *twist* up to 1.5 kb upstream of the transcription start site (Figs 7A–7F and S9). We expressed MMP28-GFP and performed the immunoprecipitation using an anti-GFP antibody (see Materials and methods). To detect potential nonspecific interactions due to the GFP tag, we performed the ChIP-PCR from embryos expressing only GFP. As a positive control for the procedure, we used Twist-GFP. GFP showed very little nonspecific binding to any regions of the different promoters (Fig 7A–7F, green dots). By contrast, pull-down with Twist (Fig 7A–7F, black dots) enriched several of the tested regions of the 3 promoters. Pull-down with MMP28-GFP led to an enrichment of 1 site on the *cad11* promoter (Fig 7C and 7D, magenta dots) and the *twist* promoter (Fig 7E and 7F, magenta dots) while no enrichment was found on *sox10* (Fig 7A and 7B, magenta dots).

Next, given that other MMPs have been found to play some transcriptional roles [14,16,35], we asked whether the interactions with *cad11* and *twist* promoters were specific to MMP28 or might be due to a general ability of MMPs to interact with chromatin. During EMT and migration, *Xenopus* neural crest express MMP14 [36] that has previously been found to display some transcriptional roles in other cell types [14]. Using fractionation and western blot, we confirmed that MMP14 is also imported in the nucleus of *Xenopus* cells (S10 Fig), making it a relevant MMP to compare with MMP28 in our ChIP-PCR experiment. Importantly, MMP14 showed no affinity to *sox10* and *twist* promoter's regions and a weak affinity to one site of the *cad11* promoter compared to MMP28 (Fig 7A–7F, black crosses). These data strongly suggest that the MMP28-promoter interactions identified here are specific to MMP28 and do not correspond to a general ability of MMPs to interact with chromatin.

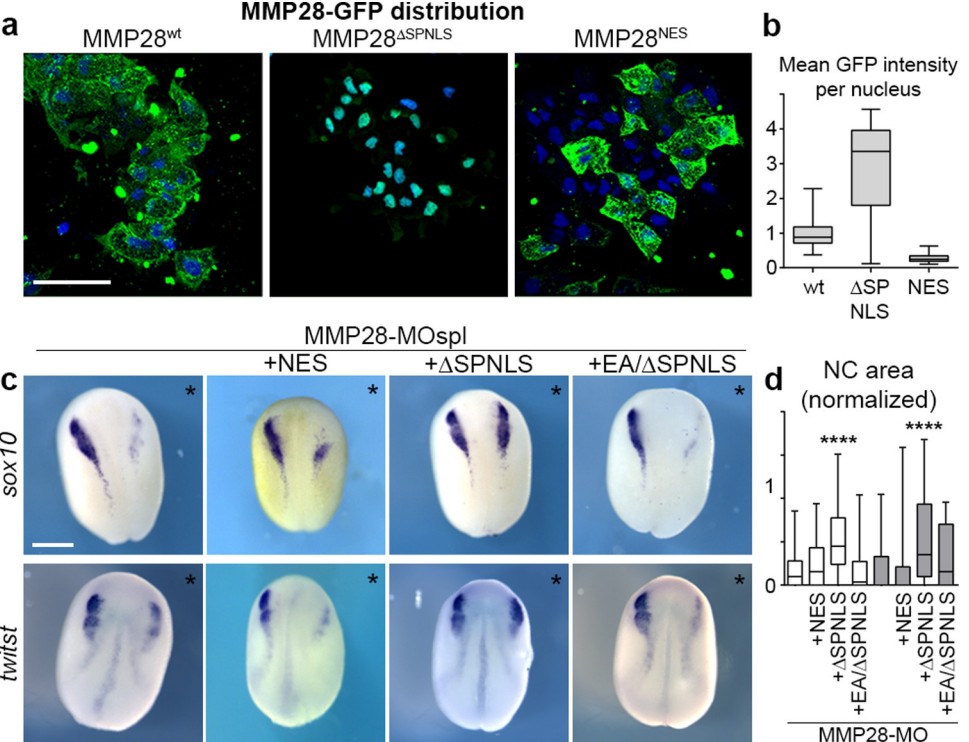

**Fig 6. Active nuclear MMP28 is required for the expression of *twist*.** (a) Neural crest expressing GFP-tagged versions of MMP28wt, MMP28$^{\Delta SPNLS}$, and MMP28$^{NES}$ cultured on Fibronectin, scale bar, 100 μm. (b) Mean intensity of the GFP signal per nucleus from 3D confocal stacks normalised to mean intensity of MMP28-GFPwt, WT ($n = 85$) from 3 explants, ΔSPNLS ($n = 39$) from 2 explants, NES ($n = 206$) from 2 explants. (c) Phenotype of MMP28-MO injected embryos co-injected with MMP28$^{NES}$-GFP, MMP28$^{\Delta SPNLS}$-GFP, or MMP28$^{EA/SPNLS}$-GFP and analysed for *twist* and *sox10* expression, scale bar, 250 μm. Asterisks mark injected side. (d) Area of neural crest gene expression normalised to non-injected side from 7 independent experiments. Number of embryos per condition, from left to right: 137, 47, 94, 28, 60, 49, 70, and 18. ANOVA followed by multiple comparisons, **** $p < 0.0001$. Numerical data from all graphs can be found in the supporting S1 Data file. MMP, matrix metalloproteinase; NC, neural crest; NES, nuclear export signal.

We showed that MMP28 knockdown leads to a reduction of *twist* and *cadherin-11* expressions assessed by qPCR (Fig 2G) and we found interactions between Twist-GFP and the *cad11* promoter region. Thus, we wondered whether Twist might be capable of inducing Cadherin-11. To test this idea, we assessed Cadherin-11 in embryos injected with CMO, MMP28-MOspl, and in the rescue condition with MMP28-MOspl with *twist* mRNA (S11 Fig). As expected, embryos injected with MMP28-MOspl had a reduction of *cadherin-11* expression compared to control MO injected ones. By contrast, in embryos co-injected with MMP28-MOspl and Twist mRNA most individuals showed an increase of Cadherin-11 on the injected side and all of these embryos had ectopic expression of *cadherin-11* compared to the uninjected contralateral side (S11 Fig).

In absence of enrichment of *sox10* promoter regions by MMP28, given that Twist seems to be upstream of Cadherin-11 and that Twist can rescue MMP28 knockdown, we decided to focus further on putative interactions between MMP28 and *twist* promoter, looking at the proximal promoter region from the transcription start site to −300 bp (Figs 7J–7L and S12). Since *e-cadherin* expression was not found to be affected in MMP28 knockdown conditions (Fig 2G), we used the *e-cadherin* promoter region as a control (Figs 7G–7I and S12). We found a very modest but significant enrichment of the 3 tested proximal regions of the *twist* promoter

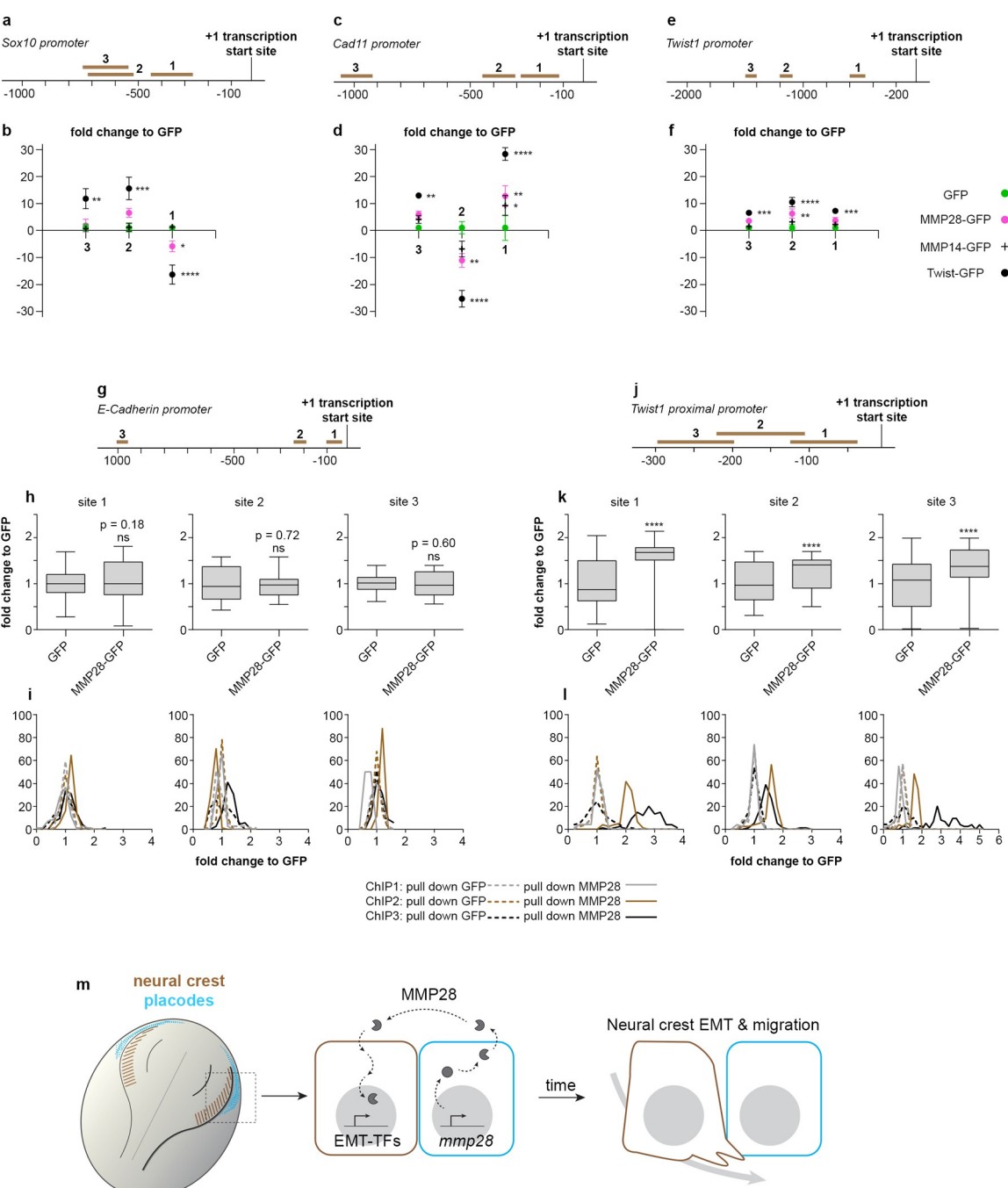

**Fig 7. MMP28 interacts with some domains of *cadherin-11* and *twist* promoters.** (a–f) ChIP-PCR after ChIP with GFP (green dots), MMP28-GFP (magenta dots), MMP14-GFP (black crosses), and Twist-GFP (black dots) for 3 domains on the *sox10* (a, b), *cadherin-11* (c, d), and *twist* (e, f) promoters, normalised to signals in GFP pull down. From 3 technical replicates, mean band intensity from gels are plotted, error bars represent the standard error of the mean. Original uncropped gels used for quantification are provided in S9 Fig. (g–l) ChIP-PCR after ChIP with GFP or MMP28-GFP for 3 domains on the *e-cadherin* (g–i) or *twist* (j–l) promoters, normalised to signals in GFP pull down. From 3 independent ChIP assays (biological replicates). Mean pixels intensity from histogram analyses from gels are shown as box and whiskers plots (h, k) or as frequency distribution per ChIP (i, l). Original uncropped gels used for quantification are provided in S12 Fig. (m) Diagram of a dorsal view of a *Xenopus* embryo with neural crest in brown and placodes in cyan. MMP28 from placodes is imported into neural crest's nuclei to promote EMT. Numerical data from all graphs can be found in the supporting S1 Data file. ChIP, chromatin immunoprecipitation; EMT, epithelial–mesenchymal transition; MMP, matrix metalloproteinase.

in the MMP28-GFP pull-down compared to pull-down with GFP alone (Fig 7J–7L). By contrast, pull-down with MMP28-GFP did not lead to any enrichment of the tested *e-cadh* promoter regions compared to GFP alone (Figs 7G–7I and S12).

It should be noted that the enrichments of promoter domains by MMP28 pull-down are of modest amplitude (1.5- to 12-fold) as compared to those observed with Twist (8- to 30-fold). In addition, MMPs in general and MMP28 in particular do not present any protein domain that could act as a direct DNA-binding domain. Further, data presented in S4 Fig show that MMP28 is not sufficient to induce *twist* expression outside of the neural crest cells by contrast with Twist's ability to induce ectopic *cadherin-11* expression (S11 Fig). These ChIP data do not imply that MMP28 is acting as a potential transcription factor. Rather, a more accurate view is that of a modulator of transcriptional activity most likely mediated via interaction with neural crest-specific co-factors involved in transcription.

In conclusion, our data indicate that active MMP28 produced by placodal cells needs to be imported into the nucleus of neural crest cells for normal Twist-dependent EMT and migration to occur (Fig 7M). These results demonstrate that MMPs are not restricted to late phases of migration and invasion but can act as upstream regulators of EMT in vivo. Given that many MMPs are expressed in the context of EMT and migration in physiological and pathological situations, it will be interesting to assess whether such paracrine role also occur in other settings.

## Discussion

We found that MMP28 is produced by cranial placodes but acts on neural crest cells. Interestingly, these 2 cell populations jointly form the cranial peripheral nervous system [37] and are known to depend on one another for their normal development. Normal neural crest migration along the dorsoventral axis and patterning in discrete streams require interactions with placodes on multiple levels. Placodes are the source of positive and negative regulators of neural crest migration such as Sdf1/CXCL12 and class3-semaphorins [38–40]. Direct physical interaction between neural crest and placode cells via N-cadherin-dependent heterotypical contact-inhibition of locomotion drives the redistribution of placode cells throughout the head which in turn helps splitting the neural crest population into streams [26,41]. These neural crest–placodes interactions are not specific to *Xenopus* development and are found in other vertebrates with some degree of conservation in terms of molecular and cellular mechanisms [42–45]. MMP28 acts earlier than the previously identified mechanisms linking neural crest and placodes development. While physical interactions and secreted guidance cues directly influence cell motility and adhesion during migration, MMP28 expression in placode cells starts around stage 14 which is after the expression of *snai2*, *sox8*, *foxd3*, and *twist* and concomitant with that of *sox10*. All of which are decreased in the absence of MMP28. It should be noted however that other neural crest genes such as *snai1* and *sox9* were not affected by MMP28 knockdown. This shows that the neural crest territory is induced and still identifiable in absence of MMP28 and that MMP28 only affects a subset of the neural crest gene regulatory network. These data indicate a new level of interaction between these 2 cell populations such that the complete neural crest EMT program is maintained only if MMP28 is secreted from the placode cells. In addition, the fact that MMP28 knockdown affects *twist*, *snai2*, and *foxd3* expression but not that of *snai1* and *sox9* echoes previous works on 2 factors, RuvB-like 1 and Prohibitin1 [46,47]. These factors have been shown to act in *Xenopus* neural crest cells as part of a signalling pathway involving c-Myc and E2F1 that is upstream of *twist*, *snai2*, and *foxd3* expression but not that of *snai1* and *sox9* suggesting that MMP28 might, in some way, be part of such pathway.

In addition to its effect on neural crest genes, we also noted a decrease of *six1* expression in the pre-placodal region after MMP28 knockdown (S2 Fig). We also showed that placodes cells can import MMP28 produced by other placode cells and that some of it is found in the nucleus (S7 Fig). Thus, one could propose that the role of MMP28 in neural crest cells is indirect, mediated by its impact on placode development via the reduction of *six1* expression. However, it seems very unlikely, as Six1 knockdown is typically associated with an expansion of *sox2* and *pax3* expression domains, and a reduction of *eya1* expression [48–50]. None of these effects were observed upon MMP28 knockdown. Further, we have shown that *six1* expression in the pre-placodal domain was not a prerequisite for that of *sox10* in the neural crest in MMP28 rescue conditions (S5 Fig). Therefore, the neural crest phenotypes observed after inhibiting MMP28 expression cannot be explained by the reduction of *six1* expression in the pre-placodal region.

We found that MMP28 is imported as an active enzyme in the nuclei of neural crest cells. How is this possible? First of all, more than half of all MMPs have been detected in the nucleus of at least 1 cell type [13], including MMP28 [18], indicating that the ability to traffic to the nucleus is a conserved property of MMPs. Putative NLSs have been found in MMP3 and its ability to go from the extracellular space to the nucleus to participate in transcription has been shown in cell culture [15]. MMP14 (a.k.a. MT1-MMP) can go to the nucleus and contributes to the transcriptional activation of PI3K in cultured mouse primary macrophages [14]. In this study, the authors showed that the transmembrane domain of MMP14 is needed for nuclear import to occur. However, a deletion construct lacking both the secretion signal and the transmembrane domain produced directly in the cytoplasm can still be imported into the nucleus. Further, direct DNA binding to the promoter of PI3K was demonstrated as well as the ability of MMP14 to activate a minimal promoter sequence. Unfortunately, the mechanisms and cofactors involved in controlling entry of MMPs into the cell and their subsequent nuclear import are still unknown and will require extensive work to be elucidated.

MMP28 expression has been described during wound healing [51]. Interestingly, its expression is not found in front cells located near the wound but in proliferative cells located several rows behind the gap that needs to be filled and does not overlap with other proteases [52]. It would be interesting to assess whether MMP28 acts in a paracrine manner in this context as well, similarly to what we found in neural crest cells. In cancer, MMP28 is detected in numerous tumour types. MMP28 promotes cell proliferation in oral squamous cell carcinoma [53], while in gastric cancer [19] and hepatocellular carcinoma (HCC) [54], there is a clear association with poor prognosis. In HCC cell lines, MMP28 overexpression promotes EMT via activation of *zeb1* and *zeb2* downstream of the Notch pathway [54]. Given that MMP28 was previously shown to favour EMT via an extracellular effect on the TGFβ pathway in lung carcinoma cell lines [9], the authors of the HCC study did not investigate whether MMP28 could act directly from within the cells.

We found MMP28 to be required for normal expression of *sox10*, *twist*, and *cad11*. Our results show that Cadherin11 and, in particular, Twist are important downstream players since their forced expression can partially (Cad11) or fully (Twist) compensate for the lack of MMP28. This is substantiated by the fact that we found MMP28 to be enriched in some domains of the promoters of these genes. The role of Sox10 in this context is less clear. Indeed, previous studies on the role of Sox10 in *Xenopus* neural crest cells did not find a role in motility but rather showed its importance in lineage choice [55,56]. This suggests that, in addition to its importance for EMT and migration, MMP28 might be required for other aspects of neural crest development via the maintenance of Sox10.

In conclusion, our results, in the context of (i) the conserved ability of MMPs to traffic to the nucleus; (ii) the few examples of interaction with some promoter regions previously

described; and (iii) the frequent expression of MMPs in cells undergoing EMT, strongly suggest that the role of MMPs as upstream regulators of EMT might be a conserved feature and would need to be systematically assessed in physiological and pathological situations.

## Materials and methods

### Ethics statement

All experiments using *Xenopus laevis* were performed in accordance with the guidelines of the Guide for the Care and Use of Laboratory Animals of the National Institutes of Health and were approved by the Institutional Animal Care and Use Committee of New York University (animal protocol # IA16-00052) or institutional and French national guidelines, under the institutional licence number A 31 55501 delivered by the Préfecture de la Haute-Garonne.

### *Xenopus* manipulation and in vitro fertilization

Female *Xenopus* laevis were injected with 750 to 1,000 International Units of Chorionic Gonadotrophin (hCG, Chorulon) and kept overnight at 18°C. Male *Xenopus laevis* were euthanized in 3 g/L Tricaine (Millipore Sigma E10521) and the testis immediately collected and kept in L15 medium (Millipore Sigma L5520) at 4°C. For fertilization, a suspension of minced testis was added to the oocytes collected in petri dish in 0.1× Normal Amphibian Medium (NAM): NaCl (110mM), KCl (2mM), Ca(CO3)2 (1mM), MgSO4 (1mM), EDTA (0.1mM), NaHCO3 (1mM), and Sodium Phosphate (2 mM).

### Expression vectors and Morpholinos for MMP28

MMP28pCMV-SPORT6 clone was purchased from OpenBiosystems (Horizon Discovery/ Dharmacon, #MXL1736-92024189). MMP28 open reading frame was amplified by PCR using the following primers, MMP28_pCS2_fdw: ATCGATATGGAAGCTGATATTCCATC MMP28_pCS2+_rev: CTCGAGTCAAGTCACATCATTTTTACA, and cloned into a pCS2+ backbone. MMP28$^{wt}$-GFP was produced from MMP28$^{wt}$-pCS2 by adding a GFP sequence using a PCR strategy with primers eGFP_pCS2_fdw: 5′– AAGGATGCAACTAGGATCCGCTCGATGAGCAAGGGCG–′3, eGFP_pCS2_rev: 5′– CGACTCACTATAGTTCTAGACTTACTTGTA–3′ and BamHI/XbaI cloning. MMP28$^{ΔSP/NLS}$- GFP-pCS2 was synthetized by Genescript. MMP28$^{ΔSP/wt}$-GFP-pCS2 was produced from MMP28$^{ΔSP/NLS}$-GFP-pCS2 by excision of the NLS sequence (BamHI/XhoI). MMP28$^{NES2}$-GFP was derived from MMP28$^{wt}$-GFP by insertion in SpeI site of the sequence 5′– CTGGCCCTGAAGCTGGCCGGCCTGGACATCGGCAGC–3′ using oligo annealing procedure. The catalytically dead mutants MMP28$^{EA}$-pCS2 and MMP28$^{EA-ΔSP/NLS}$-GFP-pCS2 were produced by point mutation of Glutamic acid$^{226}$ to Alanine from MMP28$^{wt}$pCS2 and Glutamic acid$^{209}$ to Alanine from MMP28$^{ΔSP/NLS}$-GFP-pCS2 using QuikChange II site-Directed Mutagenesis Kit (Agilent #200523) with the primers MMP28_E-A_fdw: 5′– ACTGGCACATGCGATTGGACAT–3′ and MMP28_E-A_rev: 5′– ATGTCCAATCGCATGTGCCAGT–3′.

For the secreted GFP constructs, we inserted the signal peptide of MMP28 into the construct eGFPpCS2 in NcoI restriction sites using the Plasmid Modification by Annealed Oligo Cloning method from addgene (https://www.addgene.org/protocols/annealed-oligo-cloning/) with the following primers:

    5′–CATGGAAGCTGCTATTCCATCCCTGTTCTTTCTGCTTGTGATTGCTGGTTTGTG
    CTTGC–3′ and 5′CATGGAAGCACAAACCAGCAATCACAAGCAGAAAGAACAGGGATG
    GAATAGCAGCTTC–3′.

Morpholino antisense oligonucleotides were purchased from Gene-Tools (Pilomath, Oregon, United States of America). Please note that MOatg described hereafter targets a domain upstream of ATG that is not present in the MMP28wt construct used for rescue experiments. MMP28-MOspl: 5′– GTATGCCTCTGATATTTACCTGTGC–3′, MMP28-MOatg: 5′– TGTTTAATGGATGAGTAACTTATCT–3′ and Control MO (CMO): 5′– CCTCTTACCTCAGTTACAATTTATA–3′.

## Microinjections

mRNAs were synthesised in vitro from the various pCS2 constructs using the Ambion Message Machine kit (Austin, Texas, USA). MMP28$^{wt}$ and MMP28$^{EA}$ mRNAs were injected at 8-cell stage in 2 animal blastomeres with 600 pg/bl, while all MMP28-GFP mRNAs were injected at 900 pg/bl to respect equimolarity. CMO and MMP28-MO (atg and spl) were injected at 4 ng per blastomere either alone or in combination with MMP28 mRNAs at 8-cell stage, in 2 animal blastomeres for in situ hybridization (ISH), neural crest extraction and grafting, or 4 animal blastomeres for PCR and qPCR. Nuclear mCherry [26] was injected at 10 pg per blastomere. *Twist* and *cadherin-11* mRNA were injected at 125 pg per blastomere.

## TUNEL assay

TUNEL staining was carried out as described [57]. Morpholinos-injected albinos embryos fixed in MEMFA were rehydrated in PBT and washed in TdT buffer (Invitrogen) for 30 min. End labelling was carried out overnight at room temperature in TdT buffer containing 0.5 μm DIG-dUTP and 150 U/ml TdT (Invitrogen). Embryos were then washed for 2 h at 65°C in PBS/1 mM EDTA. DIG was detected with anti-DIG Fab fragments conjugated to alkaline phosphatase (Roche, Indianapolis, Indiana, USA; 1:2,000) and the chromogenic reaction performed using BM purple (Roche, Indianapolis, Indiana, USA). Injection of a Morpholino against Sf3b4 was used as a positive control for induction of cell death [27]. For each MO, a subset of injected embryos injected was processed for in situ hybridization against sox10 as internal control for the effect of the MOs. TUNEL dots were counted in the neural crest region on each side. The neural crest region was defined as the lateral part of the anterior neural fold. Differences between injected and uninjected sides were plotted as well as the frequency distribution of TUNEL dots on the injected side for each condition.

## In situ hybridization

Embryos were fixed overnight at 4°C in MEMFA and dehydrated by several washes in methanol. Embryos were then rehydrated by solutions of decreasing methanol concentration, washed in PBS, and bleached in hydrogen peroxide (10%) to attenuate the ectoderm pigmentation. After bleaching, a short post-fixation in formaldehyde 3.7% was performed. Embryos were then processed using the InsituPro VS (Intavis AG Bioanalystical Instruments, Germany) automate. Briefly, embryos were incubated 16 h at 65°C in formamide-based hybridization buffer containing a digoxigenin or Fluorescein-12-labelled antisense probe against the gene of interest. Probes were washed in formamide-based washing solutions, then washed in PBS plus 0.1% tween, and sequentially incubated for 1 h in a serum-based blocking solution and then for 1 h in blocking solution containing the anti-digoxigenin (Roche, 11093274910; 1/2,000) or anti-Fluorescein (Roche, 11426346910; 1/10,000) antibody coupled with alkaline phosphatase. Staining was performed by incubating embryos in staining buffer (pH 9.5) containing NBT (Promega, S380C) at 50 μg/mL and BCIP (Promega, S381C) at 100 μg/mL or Magenta-Phos (Biosynth, B-7452). The following probes were used: *XL-snai2* [58], *XL-twist1* [59],

*XL-MMP28* [25], *XL-foxD3* [60], *XL-snai1* [61], *XL-sox8* [62], *XL-sox9* [63], *XL-sox10* [55], *XL-eya1* [64], XL-six1 [65], *XL-foxi4.1* [66], *XL-sox2* [67], *XL-keratin* [68], and *XL-pax3* [69].

### Ex vivo neural crest culture

Neural crest cultures were performed as described elsewhere [28]. Briefly, neural crest cells were isolated from stage 18 embryos using a hair knife (hair mounted on a glass pipette) and cut in small pieces before being plated on fibronectin-coated Ibidi μ-slides dishes (Ibidi, ref 80821). Dishes were prepared by incubating fibronectin solution at 10 μg/mL for 1 h at 37°C.

### Manual tracking

Individual neural crest cells located at the edge of explants were tracked using Manual Tracking plug-in in FIJI. Tracks were analysed using Chemotaxis Tool plug-in from IBIDI to retrieve individual cell's speed.

### Grafts and animal cap sandwiches

Grafts of neural crest cells were performed as described elsewhere [38]. Embryos at stage 18 were immobilized into a Petri dish filled with modelling clay. The pigmented ectoderm layer located above the neural crest region was carefully removed. Neural crest cells were mechanically detached from their surrounding tissues by applying gentle pressure on the side of the neural crest domain in a lateral to medial direction. To perform the graft, a given neural crest explant was first dissected out from a host embryo, then a neural crest explant was harvested from the donor embryo and grafted into the wound of the host and kept in place by a piece of glass coverslip for 15 min. The coverslip was then removed and embryos allowed to heal. For ectopic grafts of neural crest cells next to the caudal ectoderm, the region receiving the graft was first wounded by a superficial cut using a hair knife and the grafts was maintained by a glass coverslip until healing. Grafts of placodes were performed as previously described [26]. Embryos at stage 18 were placed ventral side up. Two cuts are made from dorsal to ventral, one starting from the posterior region of the neural crest domain another from the anterior neural crest domain to make a triangular cut. The whole piece of tissues containing endoderm, mesoderm, and ectoderm is lifted up to expose the cavity of the embryo without detaching the piece from the embryo. The inner layers (endoderm and mesoderm) are gently removed mechanically using a hair knife. Then, the inner layer of the ectoderm containing the placodes, lining the neural crest domain, is carefully detached from the pigmented layer and used for grafting. The neural crest cells of the host embryos are removed and replaced by placodes. This way, grafted placodes are next to placodes producing MMP28-GFP. For animal caps sandwiches, the animal poles of embryos at stage 9/10 were dissected and re-assembled ex vivo. For each sandwich, 1 large piece was dissected from an embryo expressing MMP28-GFP and 1 small piece was taken from a control embryo injected with rhodamine-dextran and separated from the superficial pigmented layer. Then, the small piece of animal cap is placed on top of the large one and the 2 tissues are left to fuse. The large piece, which still has its pigmented layer, rapidly folds on itself and engulfs the small explant.

### Immunostaining on cell cultures

*Xenopus* neural crest cells were cultured on fibronectin-coated dishes and were allowed to migrate for a few hours, fixed in 4% PFA for 30 min, blocked and permeabilized (PBS1X, 2% serum and 0.1% Triton) for 30 min and incubated 2 h at room temperature or overnight at 4°C with a primary antibody. After several washes in PBS, they were incubated 1 h at room

temperature or overnight at 4°C with a secondary antibody mixed with DAPI and/or Phalloidin, and analysed on a Zeiss 710 inverted confocal microscope. Primary antibodies: Rabbit anti-GFP (Torrey Pines BioLabs, TP-401; 1:200), Rabbit anti-Flag (MilliporeSigma, F1804; 1:200). Secondary antibodies: goat anti-rabbit Alexa-488 or 555 (Thermo Fisher Scientific).

## qPCR and PCR

Total RNAs from neural crest explants from injected embryos were extracted with RNeasy Micro Kit (Qiagen) and used for relative quantitative PCR (QuantStudio 3 real Time PCR System; Thermo Fisher Scientific) using Power SYBR Green RNA-to-$C_T$ 1-Step Kit according to the manufacturer instructions. The following primers set were used:

Cadherin-11_rev: 5′–CATCCTCTGGGTTGATGCTG 3′,
Cadherin-11_fwd: 5′–TCGGATACTGTGGTCGGAAG–3′,
N-cadherin_rev: 5′–ATTGTAACGGAGACGGTTGC–3′,
N-cadherin_fwd: 5′–CAGCAACGATGGCTTAGTGA–3′,
XB-cadherin_fwd: 5′–TATCCTTGCTGCTGCTCCTG–3′,
XB-cadherin_rev: 5′–TCACCTCCACCTTCCTCTCC–3′,
E-cadherin_rev: GCACAGAGCCTTCAAAGACC–3′,
E-cadherin_fwd: 5′–CGACCTTTGGACAGAGAAGC–3′,
EeF1a1_rev: 5′–CACGGGTTTGTCCATTCTTT–3′,
EeF1a1_fdw: 5′–ATTGATGCTCCAGGACACAG–3′,
Twist1qPCR_fdw: 5′–CGACTTTCTCTGCCAGGTCT–3′,
Twist1qPCR_rev: 5′–TCCACACGGAGAAGGCATAG–3′,
MMP28-E7_fdw: 5′–TGCAGTGGTATCGGGTTTAG–3′,
MMP28-E8_rev: 5′–AAAGTGCAGTGTCAGGACGA–3′,
Sox10_fdw: 5′–CTGTGAACACAGCATGCAAA–3′
Sox10_rev: 5′–TGGCCAACTGACCATGTAAA–3′,
MMP28-E1_fdw: 5′–GGAAGCTGCTATTCCATCCCTGT–3′,
MMP28-E1_rev: 5′–ACCTGTGCAGTTTGTAGGGTCT–3′,
Six1_fdw: 5′–CTGGAGAGCCACCAGTTCTC–3′,
Six1_rev: 5′–AGTGGTCTCCCCCTCAGTTT–3′.

All pairs of primers were validated using the standard curve method, and the data were normalised to eef1α.

For Morpholino validation by RT-PCR, total RNAs from injected embryos were extracted with RNeasy Micro Kit (Qiagen) and reverse-transcribed using SuperScript IV VILO Master Mix (Thermo Fisher Scientific, #11756050) according to the manufacturer instructions and used for PCR with Illustra PuReTaq Ready-To-Go PCR beads. The following primer sets were used:

Ornithine Decarboxylase, ODC_fdw: 5′–ACATGGCATTCTCCCTGAAG–3′,
Ornithine Decarboxylase, ODC_rev: 5′–TGGTCCCAAGGCTAAAGTTG–3′,
MMP28-E1b_fdw: 5′–GAAGAGACCCTACAAACTGC–3′,
MMP28-E8_rev: 5′–AAAGTGCAGTGTCAGGACGA–3′,
MMP28-I1-rev: 5′–GACAACGCATTTCCCAAACT–3′.

## Chromatin immunoprecipitation (ChIP)

For ChIP, we followed standard procedures established for *X. laevis* embryos [70,71]. For each independent experiment, we used 2 technical replicas and 250–300 *Xenopus* embryos per condition. Briefly, stage 18 *X. laevis* embryos were fixed for 30 min at room temperature and sonicated for 12 min at 60% power (30 s On 30 s Off) in a QSonica Q800R3 sonicator. Then, 2 μg

of Anti-GFP ChIP grade antibody (Abcam, ab290) were used to precipitate a fraction of the total sonicated extract that was equivalent to 12 embryos. For DNA extraction, we followed a standard protocol [70,71]. Using the Xenbase genome browser resource, we searched for putative chromatin accessible regions in a region spanning about 3 kb of the putative proximal promoter of *sox10*, *cadherin-11*, and *twist1* genes. Primer pairs were designed to analyse chromatin enrichment by PCR. Primers positions and sequences are listed below. PCR was performed using the following conditions 98˚C for 10 s, 64˚C for 20 s, and 72˚C for 10 s for 35 cycles.

For ChIP-PCR with GFP, MMP28-GFP, MMP14-GFP, and TWIST-GFP on *sox10*, *cadherin-11*, and *twist* promoters:

sox10_F1: 5′–GCAATGTACCGGCTGCAATA–3′; -451
sox10_R1:5′–CGTCGCACAGTGCTTCTTT–3′; -269
sox10_F2: 5′–CTGCAACTCTCCAGCTCTTT–3′; -734
sox10_R2: 5′–GCAGTCTGTGTTAATGCAAGTC–3′; -526
sox10_F3: 5′–AATCTAGGAAAGTACGTCAGTGC–3′; -754
sox10_R3: 5′–CCACCCTCCTGGACTAATAAATG–3′; -536

cad11_F1: 5′–TCCAGTCTCTGCATCACTTTATC–3′; -284
cad11_R1: 5′–GGGTCTCTCAGCTTCTCTTTC–3′; -136
cad11_F2: 5′–TCCCACACACACACATCTTATC–3′; -444
cad11_R2: 5′–ATGGAAGCATATGGAGGAAAGG–3′; -310
cad11_F3: 5′–CCCAGACCAATAGAACCACTTT–3′; -1057
cad11_R3: 5′–GAGGGTGTTAATTGGCCCTAA–3′; -929

twist1_F1: 5′–ACAATCCGCGCTAAGTAAAGA–3′; -600
twist1_R1: 5′–GGATCCTATGGGAATGGGAAAG–3′; -481
twist1_F2: 5′–GGACCCAGTCTAAGGGAATAGA–3′; -1189
twist1_R2: 5′–TCAGCCACCCTTCACATTTAG–3′; -1109
twist1_F3: 5′–GGCTGGTACAGAAGCTCAAA–3′; -1506
twist1_R3: 5′–CCAGGACAGGCATGTGTATAG–3′; -1408

For ChIP-PCR with GFP and MMP28-GFP on *e-cadherin* and *twist* promoters:
Primers sequence and mapping relative to cadh1.S exon1
F1/R1 [-126;-9]
cadh1.S_F1 5′–CAGGGCAGAGACATTCAGTAG–3′
cadh1.S_R1 5′–TTTGGAAGGACAGAAGGATCAG–3′
F2/R2 [-276;-172]
cadh1.S_F2 5′–TGACTCACAGAGTGCCAATAC–3′
cadh1.S_R2 5′–AGTCGCTTCTCATTGGTCAG–3′
F3/R3 [-1053;-916] (overlaps putative E-Box)
cadh1.S_F3 5′–CCTTCAATTACAGGTATGGGATCT–3′
cadh1.S_R3 5′–GGCCTTACCCTAAAGGAACAA–3′
F1/R1 [-123; -31]
twist1.L_F1 5′–TCTTTCTGCCAGCGACAATA–3′
twist1.L_R1 5′–CCCAGCCCTTGTACAGTC–3′
F2/R2 [-220;-108]
twist1.L_F2 5′–TCTGACTAGCGGAAACCTAATAAA–3′
twist1.L_R2 5′–GTCGCTGGCAGAAAGATTTG–3′
F3/R3 [-298;-197]

twist1.L_F3 5′–CCTTCTTCATTCTGATGGCAAA–3′
Twist1.L_R3 5′–TTTATTAGGTTTCCGCTAGTCAG–3′.

Band intensities were measured using FIJI, either mean band intensities or pixel intensity distribution were used (see legends of Fig 7 for details). For each band, the level of background signal was measured in a portion of the gel directly adjacent to the band and subtracted. Each dataset was then normalised to the peak value and to signals in the GFP pull down conditions.

## Statistics

Comparison of percentages was performed using contingency tables [72]. Two datasets were considered significantly different (null hypothesis rejected) if $T > 3.841$ ($\alpha = 0.05$, *), $T > 6.635$ ($\alpha = 0.01$, **) or $T > 10.83$ ($\alpha = 0.001$, ***). Normality of datasets was tested using Kolmogorov–Smirnov's test, d'Agostino and Pearson's test and Shapiro–Wilk's test using Prism6 (GraphPad). A dataset was considered normal if found as normal by all 3 tests. Datasets following a normal distribution were compared with Student $T$ test (two-tailed, unequal variances) or a one-way ANOVA with multiple comparisons post-test in Prism6 (GraphPad). Datasets that did not follow a normal distribution were compared using Mann–Whitney's test or a nonparametric ANOVA using Prism6 (GraphPad). Cross-comparisons were performed only if overall P value of the ANOVA was <0.05. Strategy for sample size determination does not apply here since all embryos or cells available were analysed. Statistics were performed on the whole population. Variances were not assumed to be equal. Box and whiskers plot: the box extends from the 25th to the 75th percentile; the whiskers show the extent of the whole dataset. The median is plotted as a line inside the box. Statistics are provided in figure legends or added directly onto the graphs. All error bars on graphs and curves that are not box and whiskers plots correspond to the standard deviation (SD) or standard error of the mean (SEM) as indicated in the figure legends.

## Image analysis

The net distance of in vivo neural crest cells migration was measured using FIJI/Image J by drawing a straight line between the dorsal midline and the ventral-most neural crest cells of each stream. The mean length of dorsoventral extension is calculated per embryo and each side (experimental versus control/non-injected). The ratio of the mean dorsoventral extension is plotted after normalisation to the control condition of reference.

Neural crest areas at pre-migratory stages were measured as follows: ISH images were converted to 32-bit black and white (NBT/BCIP magenta staining being white on a black background), thresholded with background converted to NaN, automatic measurements of area were made with batch processing in FIJI/imageJ. If an embryo had too much background after ISH, the size of the NC area was retrieved by hand in FIJI/ImageJ using the polygon tool.

Analysis of nuclear localization of the various MMP28-GFP constructs was performed from 3D confocal stacks with optimal pinhole settings acquired after immunostaining against GFP and counterstaining with DAPI. In Imaris (Bitplane), the DAPI channel was used as a mask to sample the GFP channel. Then, surface analysis was performed to calculate the volume of the nuclei and the volume of the GFP within the nuclei. Nuclear localization is expressed as a ratio between the volume of nuclear GFP immunostaining and the nuclear volume.

## Bioinformatics analysis of *Xenopus* MMP28 protein sequence

*Xenopus* laevis MMP28 amino acid sequence was searched for putative importin α-dependent NLSs and NESs using cNLS mapper [73] (http://nls-mapper.iab.keio.ac.jp/cgi-bin/NLS_Mapper_form.cgi) and ValidNESs [74] (http://validness.ym.edu.tw/).

## Western blots

Cell fractions of 50 (100 mg) stage 17 wild-type or injected embryos were obtained using the Subcellular Protein Fractionation Kit for Tissues (Thermo Fisher Scientific # 87790). Protein quantification was performed using Bradford technique (Pierce). Around 40 μg of proteins were loaded per lane of a 4% to 12% precast gel (Bio-Rad Mini-PROTEAN). Proteins were transferred on PVDF-membrane and blocked for 1 h with 5% skimmed milk. The following antibodies were used: anti-GFP 1:1,000 (Torrey Pines BioLabs; TP-401), anti-Tubulin 1:1,000 (Sigma, T9026), anti-rabbit-HRP, or anti-mouse-HRP 1:10,000 (Millipore).

The numerical data used in all figures are included in S1 Data.

## Illustrations

All drawings in figures and supporting files were generated by ET using Adobe Illustrator CS6.

## Supporting information

**S1 Fig. Effect of broad or targeted MMP28 knockdown on Sox10 expression.** (a, b) Representative images of embryos injected with MMP28-MOspl in a broad manner (a, $n = 15$), overlapping with the placodal domain or in a targeted manner (b, $n = 15$) targeting the neural fold and medial crest subregion but excluding the placodes, analysed by in situ hybridization against Sox10. MO was co-injected with rhodamine-dextran for tracing (red).
(TIF)

**S2 Fig. Effect of MMP28 knockdown on ectoderm patterning.** (a) Diagram representing the anterior view of Xenopus laevis neurula (Stage 16). Embryos were injected on their left-hand side (green). (b) Representative images of embryos injected with either control MO (CMO) or MMP28-MOspl after in situ hybridization for cytokeratin, pax3, sox2, eya1, foxi4.1 and six1, asterisks mark the injected side. (c) Ratio of the width of the cytokeratin-negative domain from the midline for the injected and uninjected sides, normalised to the CMO condition; CMO ($n = 43$), MMP28MOspl ($n = 63$). (d) Percentages of embryos with normal or affected expression of pax3 after injection of CMO ($n = 40$) or MMP28-MOspl ($n = 20$). (e) Ratio of the width of the sox2-positive domain from the midline for the injected and uninjected sides, normalised to the CMO condition; (CMO $n = 20$), MMP28MOspl ($n = 15$). (f) Ratio of area of gene expression for the injected and uninjected sides, normalised to the CMO condition; eya1: (CMO $n = 15$), MMP28MOspl ($n = 14$); foxi4.1: CMO ($n = 27$), MMP28MOspl ($n = 37$); six1: CMO ($n = 20$), MMP28MOspl ($n = 49$). Statistics: Student $t$ tests c, $p = 0.3510$; e, $p = 0.7125$; f, eya1, $p = 0.3572$; foxi4.1, $p = 0.3512$; six1, $p < 0.0001$. For panel (d), contingency table T = 0.135, $\alpha$ = ns. Scale bar, 200 μm.
(TIF)

**S3 Fig. MMP28 loss-of-function does not lead to cell death in the neural crest domain.** (a) Representative images of a TUNEL assay in embryos injected with 10 ng of Morpholino against Sf3b4 (used as a positive control for induction of cell death, see Materials and methods), 4 ng of control MO or 4 ng of MMP28spl-MO in 2 blastomeres at 8-cell stage. (b) Differences between the number of TUNEL dots on each side within the neural crest region. The neural crest region was defined as the lateral half of each anterior neural fold. A value of 0 means that each side had the same number of TUNEL dots on either side. A negative value indicates more cell death on the non-injected side than the injected side while a positive value indicates more cell death on the injected side. (c) Frequency distribution of TUNEL dots in the neural crest region on the injected side in all conditions. Respectively 96% and 94% of all embryos injected with CMO or MMP28spl-MO only had between 0 and 20 TUNEL dots in

the injected neural crest region, the remaining 4/6% had more than 20 TUNEL dots. By contrast, after inhibition of Sf3b4, only 37% of embryos had between 0 and 20 TUNEL dots in injected neural crest cells, 47% had more than 20 dots, and the remaining 8% had 30 dots or more on the injected side. ANOVA, Kruskal–Wallis; **** $p < 0.0001$.
(TIF)

**S4 Fig. In vivo overexpression of MMP28wt-GFP and MMP28-EA-GFP.** (a, b) In situ hybridization for Twist in uninjected embryos (a) and embryos injected with 900 pg MMP28wt-GFP or MMP28-EA-GFP mRNA (b). (c) Mean dorsoventral migration of neural crest cells in uninjected controls or after MMP28 overexpression, normalised to control side. ANOVA, followed by multiple comparisons. Uninjected vs. MMP28wt-GFP, $p = 0.132$ (ns); uninjected vs. MMP28-EA-GFP, $p = 0.699$ (ns); MMP28wt-GFP vs. MMP28-EA-GFP, $p = 0.189$ (ns). Note that MMP28 overexpression does not affect neural crest migration and does not induce ectopic Twist expression.
(TIF)

**S5 Fig. Rescue of Sox10 expression can occur in absence of a rescue of Six1 expression.** (a–f) In situ hybridization for Sox10 (a–c) and Six1 (d–f), in embryos injected with CMO (a, $n = 29$; d, $n = 56$), MMP28-MOspl (b, $n = 40$; e, $n = 43$), or co-injected with MMP28-MOspl and mRNA for MMP28wt-GFP (c, $n = 31$; f, $n = 41$). (g, h) Proportions of embryos with symmetrical or decreased expression of Sox10 (g) or Six1 (h) in each experimental condition. Contingency tables for comparison of proportions: Sox10 CMO vs. MOspl, T = 57.34 (***), MOspl vs. rescue condition, T = 36.65 (***); Six1 CMO vs. MOspl, T = 70.59 (***), MOspl vs. rescue condition, T = 7.7 (***), Rescue Sox10 vs. Rescue Six1, T = 19.5 (***). Asterisks on images indicate the injected side. Dotted lines mark the midline of each embryo. Brown arrows indicate the missing portion of Six1 expression domain in MMP28-MOspl and MMP28-MOspl +mRNA MMP28wt-GFP embryos. Scale, embryos are 500 μm wide on average.
(TIF)

**S6 Fig. MMP28 can traffic from the ectoderm to the nuclei of neural crest cells in vivo.** (a) Immunostaining against-GFP on cryosections of embryos expression MMP28wt-GFP in which rhodamine-dextran (RDx) positive control neural crest cells were grafted. (b) Immunostaining against-GFP on cryosections of non-injected embryos (negative controls) in which rhodamine-dextran positive neural crest cells were grafted. Nuclei were counterstained with DAPI (magenta); immunostaining for GFP is shown in green and rhodamine-dextran in grey. Scale bars, panel (a) 200 μm (low magnification), 20 μm (high magnification), and 5 μm on zooms. Note that in absence of GFP, the GFP immunostaining gives no significant signal.
(TIF)

**S7 Fig. Paracrine import of MMP28-GFP is not restricted to neural crest placodes interactions.** (a, b) Graft of neural crest cells labelled with rhodamine-dextran next to caudal ectoderm expressing MMP28-GFP. (c, d) Graft of placodals cells labelled with rhodamine-dextran next to placodal cells expressing MMP28-GFP. (e, f) Animal cap sandwiches between animal caps expressing MMP28-GFP and animal caps labelled with rhodamine-dextran. Counterstaining with DAPI (magenta). Rhodamine-dextran (grey), MMP28-GFP (green). No anti-GFP staining was performed on these samples. Scale is given by the nuclei that have a diameter of 12 μm on average. Numbers and quantifications of internalisation and nuclear import are shown on Fig 5, panel (c).
(TIF)

**S8 Fig. Processing of MMP28 requires entry into the secretion pathway.** (a, b) Western blots using anti-GFP antibody after cell fractionation from embryos expressing MMP28-GFP (WT), MMP28-ΔSP (deletion of the secretion peptide), or MMP28-ΔSPNLS (deletion of the secretion peptide and insertion of a strong NLS in C-terminus) on the soluble (a) and chromatin-associated (b) nuclear fractions, representative image from 2 independent experiments. Lamin B1 and Histone H3 were used as controls for the soluble and chromatin-associated fractions, respectively. Note that the lower band of MMP28 (circa 70kDa) is not detected in the ΔSP and ΔSPNLS conditions indicating that the pro-domain of MMP28 is not removed if MMP28 is prevented from entering the secretion pathway.
(TIF)

**S9 Fig. PCR after chromatin immunoprecipitation with GFP, MMP14-GFP, MMP28-GFP, and Twist-GFP.**
(TIF)

**S10 Fig. Cell fractionation after expression of MMP14-GFP.** (a–b') Western blots using anti-GFP antibody after cell fractionation from embryos expressing MMP14-GFP; (a) and (b) are 2 fractionations from independent samples, and (a') and (b') are the same blots as (a) and (b) with exposure time optimised for band detection in the nuclear fractions.
(TIF)

**S11 Fig. Twist is sufficient to drive ectopic Cadherin-11 expression.** (a, b) Embryos injected with control MO. (c, d) Embryos injected with MMP-MOspl. (c–f) Embryos injected with MMP28-MOspl and Twist mRNA. (g) Dorsal view of the embryo shown in (e) and (f). (h) Percentage of embryos with expression of Cadherin-11 that is either symmetrical (grey) or increased/reduced (brown/black) on the injected side. Note that all embryos that had an increased expression of Cadherin-11 in the MMP28-MOspl+Twist mRNA condition also had ectopic expression of Cadherin-11 in the ectoderm as seen on panel (g) (arrows). Asterisk indicates the injected side.
(TIF)

**S12 Fig. PCR after chromatin immunoprecipitation with GFP and MMP28-GFP.** (a, b) Original images for the 3 independent ChIP assays analysed in Fig 7G–7L. On each gel, the band located left to the marker of size is the positive control of PCR efficiency for each site on total chromatin extracts.
(TIF)

**S1 Movie. Ex vivo dispersion assay with neural crest cells from embryos injected with CMO or MMP28-MOatg.** Left panels: 2 examples of neural crest cells injected with CMO. Middle and right panels: 4 examples of neural crest cells injected with MMP28-MOatg. One image every 3 min, 10× objective. Total duration, 7.5 h. Related to Fig 2C–2E.
(AVI)

**S2 Movie. Ex vivo dispersion assay with neural crest cells from embryos injected with CMO, MMP28-MOatg with or without MMP28wt or MMP28wt overexpression.** From left to right: 2 examples of neural crest cells injected with CMO, MOatg, MOatg+MMP28wt, MMP28wt. One image every 3 min, 10× objective. Total duration, 7.5 h. Related to Fig 2C–2E.
(AVI)

**S3 Movie. CMO vs. MMP28-MOspl or MMP28-MOspl and Twist or Cadherin-11 mRNA, neural crest cells ex vivo dispersion assay.** Top left panel: neural crest cells injected with CMO. Top right panel: neural crest cells injected with MMP28-MOspl. Bottom left panel:

neural crest cells injected with MMP28-MOspl+Twist mRNA. Bottom right panel: neural crest cells injected with MMP28-MOspl+Cadherin11 mRNA. One image every 3 min, 10× objective. Total duration, 7.5 h. Related to Fig 3I–3K.
(AVI)

**S4 Movie. 3D confocal imaging of neural crest cells expressing MMP28-GFP.** Neural crest cells express MMP28-GFP (green) and are counterstained with DAPI (blue). DAPI staining is used as a mask to sample the green channel, highlighting the amount of MMP28-GFP present in the nuclei. Related to Fig 4.
(AVI)

**S5 Movie. 3D confocal imaging of neural crest cells expressing MMP28-flag.** Flag was detected by immunostaining (grey) and cells are counterstained with DAPI (blue). DAPI staining is used as a mask to sample the grey channel, highlighting the amount of MMP28-flag present in the nuclei. Related to Fig 4.
(AVI)

**S1 Data. Numerical values for all plots and graphs.**
(XLSX)

**S1 Raw Images. Original images of all blots and gels.**
(PDF)

## Acknowledgments

The authors are grateful to Profs. Marianne Bronner (CalTech) and A-H Monsoro-Burq (Institut Curie, College de France) as well as Drs. Bertrand Benazeraf (CNRS, University of Toulouse), Kyra Campbell (Sheffield University), Guojun Sheng (Kumamoto University), Sei Kuriyama (Akita University), and Ben Steventon (Cambridge University) for critical reading of the manuscript and friendly advice.

## Author Contributions

**Conceptualization:** Nadège Gouignard, Eric Theveneau.

**Data curation:** Nadège Gouignard, Anne Bibonne, Jean-Pierre Saint-Jeannet, Eric Theveneau.

**Formal analysis:** Anne Bibonne, Jean-Pierre Saint-Jeannet, Eric Theveneau.

**Funding acquisition:** Nadège Gouignard, Eric Theveneau.

**Investigation:** Nadège Gouignard, Anne Bibonne, João F. Mata, Fernanda Bajanca, Bianka Berki, Elias H. Barriga, Jean-Pierre Saint-Jeannet, Eric Theveneau.

**Methodology:** Nadège Gouignard, João F. Mata, Fernanda Bajanca, Bianka Berki, Elias H. Barriga, Jean-Pierre Saint-Jeannet, Eric Theveneau.

**Project administration:** Eric Theveneau.

**Supervision:** Elias H. Barriga, Jean-Pierre Saint-Jeannet, Eric Theveneau.

**Validation:** Eric Theveneau.

**Writing – original draft:** Eric Theveneau.

**Writing – review & editing:** Nadège Gouignard, Jean-Pierre Saint-Jeannet, Eric Theveneau.

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
