## [Editor Report · Decision Letter 0]

14 Feb 2023

Dear Dr Theveneau, 

Thank you for submitting your manuscript entitled "Paracrine regulation of neural crest EMT by placodal MMP28" for consideration as a Research Article by PLOS Biology.

Your manuscript and the Review Commons reviews have now been evaluated by the PLOS Biology editorial staff, as well as by an academic editor with relevant expertise. I am writing to let you know that we would like to send your submission out for external peer review to the original Review Commons reviewers.

IMPORTANT: Before we can send your manuscript to reviewers, we need you to complete your submission by providing the metadata that is required for full assessment. To this end, please login to Editorial Manager where you will find the paper in the 'Submissions Needing Revisions' folder on your homepage. Please click 'Revise Submission' from the Action Links and complete all additional questions in the submission questionnaire.

Once your full submission is complete, your paper will undergo a series of checks in preparation for peer review. After your manuscript has passed the checks it will be sent out for review. To provide the metadata for your submission, please Login to Editorial Manager (https://www.editorialmanager.com/pbiology) within two working days, i.e. by Feb 16 2023 11:59PM.

Kind regards,

Kris

Kris Dickson, Ph.D., (she/her)

Neurosciences Senior Editor/Section Manager

PLOS Biology

kdickson@plos.org

---

## [Decision Letter · Decision Letter 1]

27 Apr 2023

Dear Dr Theveneau,

Thank you for your patience while we considered your revised manuscript from Review Commons entitled "Paracrine regulation of neural crest EMT by placodal MMP28" for consideration as a Research Article at PLOS Biology. Your revised study has now been evaluated by the PLOS Biology editors, the Academic Editor and two of the original reviewers. 

In light of the reviews, which you will find at the end of this email, we are pleased to offer you the opportunity to address the remaining points from Reviewer 1 in a revision that we anticipate should not take you very long. We will then assess your revised manuscript and your response to the reviewers' comments with our Academic Editor aiming to avoid further rounds of peer-review, although might need to consult with the reviewers, depending on the nature of the revisions.

**IMPORTANT - SUBMITTING YOUR REVISION**

3. Resubmission Checklist

a) *PLOS Data Policy*

Please note that as a condition of publication PLOS' data policy (http://journals.plos.org/plosbiology/s/data-availability) requires that you make available all data used to draw the conclusions arrived at in your manuscript. If you have not already done so, you must include any data used in your manuscript either in appropriate repositories, within the body of the manuscript, or as supporting information (N.B. this includes any numerical values that were used to generate graphs, histograms etc.).

We need you to provide the data underlying the graphs shown in the following figures:

Fig. 1H, K; Fig. 2B, D, E, G; Fig. 3C, F, H, J, K; Fig. 5C; Fig. 6B, D; Fig. 7B, D, F, H, I, K, L; Fig. S2C-F; Fig. S3B, C; Fig. S4C; Fig. S5G, H; Fig. S11H

Please also indicate in each figure legend WHERE THE DATA CAN BE FOUND. For an example see here: http://www.plosbiology.org/article/info:doi%2F10.1371%2Fjournal.pbio.1001908#s5

b) *Published Peer Review*

d) *Blurb*

Please also provide a blurb which (if accepted) will be included in our weekly and monthly Electronic Table of Contents, sent out to readers of PLOS Biology, and may be used to promote your article in social media. The blurb should be about 30-40 words long and is subject to editorial changes. It should, without exaggeration, entice people to read your manuscript. It should not be redundant with the title and should not contain acronyms or abbreviations. For examples, view our author guidelines: https://journals.plos.org/plosbiology/s/revising-your-manuscript#loc-blurb

Sincerely,

Ines

--

Ines Alvarez-Garcia, PhD

Senior Editor

PLOS Biology

Reviewers' comments

Rev. 1: Shuyi Nie - note that this reviewer has signed her review

The authors addressed most of my previous comments and have included additional experiments to clarify their conclusions and improve the manuscript.

There is just one minor point regarding the previous comment on Figure 3j. I agree that if the time takes for explants to attach and flatten varies, then comparing to T0, which is one hour after placing the explants may lead to misleading conclusions. However, comparing the area of explants with the area of control explants at the end of the experiment has to presume all explants are of exactly the same size. However, no matter how careful the explants are dissected, the variations in explant size in inevitable, as shown by the diverse range in Figure 3k. Given this variation, the explants receiving MMP28-MOspl and Cad11 could possibly be about half the size of control explants and spread equally well as control explants as reflected by the graph. My suggestion is to start measuring explant area right after plating, and compare the area of the same explant from its new time 0, so that the dynamic change can be captured.

Rev. 2:

This report by Gouignard et al. examines the function of the matrix metalloproteinase MMP28 in regulating epithelial-to-mesenchymal transition (EMT) in the Xenopus neural crest. The authors show that MMP28 is expressed by the placodal cells, is secreted, and subsequently internalized by the adjacent neural crest. Using in vivo and in vitro perturbations, the authors demonstrate that MMP28 catalytic function is necessary for neural crest marker expression, migration, and adhesion. Further experiments convincingly demonstrate that MMP28 from the placodes is translocated into the nucleus of neural crest cells. The authors show that this phenomenon can occur between multiple sending and receiving cell populations but occurs most robustly between the placode-neural crest. Finally, rescue experiments indicate that MMP28 nuclear localization is necessary to rescue neural crest marker expression, and thorough ChIP-PCR experiments suggest direct interactions between MMP28 and neural crest genes.

This manuscript is clearly presented, and I commend the authors on the new experiments, analyses, and text revisions that have carefully addressed the points raised by myself and the two other previous reviewers from Review Commons. As such, I find this manuscript to be suitable for publication in PLOS Biology and believe it will significantly impact future EMT studies.

---

## [Editor Report · Decision Letter 2]

18 Jul 2023

Dear Dr Theveneau,

Thank you for the submission of your revised Research Article entitled "Paracrine regulation of neural crest EMT by placodal MMP28" for publication in PLOS Biology. On behalf of my colleagues and the Academic Editor, Anna Kicheva, I am delighted to let you know that we can in principle accept your manuscript for publication, provided you address any remaining formatting and reporting issues. These will be detailed in an email you should receive within 2-3 business days from our colleagues in the journal operations team; no action is required from you until then. Please note that we will not be able to formally accept your manuscript and schedule it for publication until you have completed any requested changes.

PRESS

Sincerely, 

Ines

--

Ines Alvarez-Garcia, PhD

Senior Editor

PLOS Biology
